# Stochastic Frank-Wolfe for Composite Convex Minimization

**Francesco Locatello**[*]     **Alp Yurtsever**[†]     **Olivier Fercoq**[‡]     **Volkan Cevher**[†]

francesco.locatello@inf.ethz.ch
{alp.yurtsever,volkan.cevher}@epfl.ch
olivier.fercoq@telecom-paristech.fr

[*]Department of Computer Science, ETH Zurich, Switzerland
[†]LIONS, Ecole Polytechnique Fédérale de Lausanne, Switzerland
[‡]LTCI, Télécom Paris, Université Paris-Saclay, France

## Abstract

A broad class of convex optimization problems can be formulated as a semidefinite program (SDP), minimization of a convex function over the positive-semidefinite cone subject to some affine constraints. The majority of classical SDP solvers are designed for the deterministic setting where problem data is readily available. In this setting, generalized conditional gradient methods (aka Frank-Wolfe-type methods) provide scalable solutions by leveraging the so-called linear minimization oracle instead of the projection onto the semidefinite cone. Most problems in machine learning and modern engineering applications, however, contain some degree of stochasticity. In this work, we propose the first conditional-gradient-type method for solving stochastic optimization problems under affine constraints. Our method guarantees $\mathcal{O}(k^{-1/3})$ convergence rate in expectation on the objective residual and $\mathcal{O}(k^{-5/12})$ on the feasibility gap.

## 1 Introduction

We focus on the following stochastic convex composite optimization template, which covers finite sum and online learning problems:

$$\underset{x \in \mathcal{X}}{\text{minimize}} \quad \mathbb{E}_\Omega f(x, \omega) + g(Ax) := F(x). \tag{P}$$

In this optimization template, we consider the following setting:
▷ $\mathcal{X} \subset \mathbb{R}^n$ is a convex and compact set,
▷ $\omega$ is a realization of the random variable $\Omega$ drawn from the distribution $\mathcal{P}$,
▷ $\mathbb{E}_\Omega f(\cdot, \omega) : \mathcal{X} \to \mathbb{R}$ is a smooth (see Section 1.2 for the definition) convex function,
▷ $A \in \mathbb{R}^n \to R^d$ is a given linear map,
▷ $g : \mathbb{R}^d \to \mathbb{R} \cup \{+\infty\}$ is a convex function (possibly non-smooth).

We consider two distinct specific cases for $g$:
(i) $g$ is a Lipschitz-continuous function, for which the proximal-operator is easy to compute:

$$\text{prox}_g(y) = \arg \min_{z \in \mathbb{R}^d} \ g(z) + \frac{1}{2}\|z - y\|^2 \tag{1}$$

(ii) $g$ is the indicator function of a convex set $\mathcal{K} \subset \mathbb{R}^d$:

$$g(z) = \begin{cases} 0 & \text{if } z \in \mathcal{K}, \\ +\infty & \text{otherwise.} \end{cases} \tag{2}$$

The former covers the regularized optimization problems. This type of regularization is common in machine learning applications to promote a desired structure to the solution. The latter handles affine constraints of the form $Ax \in \mathcal{K}$. We can also attack the combination of both: the minimization of a regularized loss-function subject to some affine constraints.

In this paper, we propose a conditional-gradient-type method (*aka* Frank-Wolfe-type) for (P). In summary, our main contributions are as follows:

▷ We propose the first CGM variant for solving (P). By CGM variant, we mean that our method avoids projection onto $\mathcal{X}$ and uses the *lmo* of $\mathcal{X}$ instead. The majority of the known methods for (P) require projections onto $\mathcal{X}$.

▷ We prove $\mathcal{O}(k^{-1/3})$ convergence rate on objective residual when $g$ is Lipschitz-continuous.

▷ We prove $\mathcal{O}(k^{-1/3})$ convergence rate on objective residual, and $\mathcal{O}(k^{-5/12})$ on feasibility gap when $g$ is an indicator function. Surprisingly, affine constraints that make the *lmo* challenging for existing CGM variants can be easily incorporated in this framework by using smoothing.

▷ We provide empirical evidence that validates our theoretical findings. Our results highlight the benefits of our framework against the projection-based algorithms.

## 1.1 Motivation: Stochastic Semidefinite Programming

Consider the following stochastic semidefinite programming template, minimization of a convex function over the positive-semidefinite cone subject to some affine constraints:

$$\underset{X \in \mathbb{S}^n_+, \ \operatorname{tr}(X) \leq \beta}{\text{minimize}} \quad \mathbb{E}_\Omega f(X, \omega) \quad \text{subject to} \quad AX \in \mathcal{K}. \tag{3}$$

Here, $\mathbb{S}^n_+$ denotes the positive-semidefinite cone. We are interested in solving (3) rather than the classical SDP since it does not require access to the whole data at one time. This creates a new vein of SDP applications in machine learning. Examples span online variants of clustering [33], streaming PCA [4], kernel learning [24], community detection [1], optimal power-flow [29], etc.

**Example: Clustering.** Consider the SDP formulation of the k-means clustering problem [33]:

$$\underset{X \in \mathbb{S}^n_+, \ \operatorname{tr}(X) = k}{\text{minimize}} \quad \langle D, \, X \rangle \quad \text{subject to} \quad X1_n = 1_n, \quad X \geq 0. \tag{4}$$

Here, $1_n$ denotes the vector of ones, $X \geq 0$ enforces entrywise non-negativity, and $D$ is the Euclidean distance matrix. Classical SDP solvers assume that we can access to the whole data matrix $D$ at each time instance. By considering (3), we can solve this problem using only a subset of entries of $D$ at each iteration. Remark that a subset of entries of $D$ can be computed form a subset of the datapoints, since $D$ is the Euclidean distance matrix.

We can attack (P), and (3) as a special case, by using operator splitting methods, assuming that we can efficiently project a point onto $\mathcal{X}$ (see [2] and the references therein). However, projection onto semidefinite cone might require a full eigendecomposition, which imposes a computational bottleneck (with its cubic cost) even for medium scaled problems with a few thousand dimensions.

When affine constraints are absent from the formulation (3), we can use stochastic CGM variants from the literature. The main workhorse of these methods is the so-called linear minimization oracle:

$$S = \arg\min_Y \ \left\{ \langle \nabla f(X, \omega), Y \rangle : \quad Y \in \mathbb{S}^n_+, \ \operatorname{tr}(Y) \leq \beta \right\} \tag{*lmo*}$$

We can compute $S$ if we can find an eigenvector that corresponds to the smallest eigenvalue of $\nabla f(X, \omega)$. We can compute these eigenvectors efficiently by using shifted power methods or the randomized subspace iterations [12]. When we also consider affine constraints in our problem template, however, *lmo* becomes an SDP instance in the canonical form. In this setting, neither projection nor *lmo* is easy to compute. To our knowledge, no existent CGM variant is effective for solving (3) (and (P)). We specifically bridge this gap.

## 1.2 Notation and Preliminaries

We denote the expectation with respect to the random variable $\Omega$ by $\mathbb{E}_\Omega$, and the expectation wrt the sources of randomness in the optimization simply by $\mathbb{E}$. Furthermore we denote $f^\star := \mathbb{E}_\Omega f(x^\star, \omega)$ where $x^\star$ is the solution of (P). Throughout the paper, $y^\star$ represents the solution of the dual problem of (P). We assume that strong duality holds. Slater's condition is a common sufficient condition for strong duality that implies existence of a solution of the dual problem with finite norm.

**Solution.** We denote a solution to (P) and the optimal value by $x^\star$ and $F^\star$ respectively:

$$F^\star = F(x^\star) \leq F(x), \qquad \forall x \in \mathcal{X}. \tag{5}$$

We say $x^\star_\epsilon \in \mathcal{X}$ is an $\epsilon$-suboptimal solution (or simply an $\epsilon$-solution) if and only if

$$F(x^\star_\epsilon) - F^\star \leq \epsilon. \tag{6}$$

**Stochastic first-order oracle (sfo).** For the stochastic function $\mathbb{E}_\Omega f(x, \omega)$, suppose that we have access to a stochastic first-order oracle that returns a pair $(f(x, \omega), \nabla f(x, \omega))$ given $x$, where $\omega$ is an *iid* sample from distribution $\mathcal{P}$.

**Lipschitz continuity & Smoothness.** A function $g : \mathbb{R}^d \to \mathbb{R}$ is $L$-Lipschitz continuous if

$$|g(z^1) - g(z^2)| \leq L\|z^1 - z^2\|, \qquad \forall z^1, z^2 \in \mathbb{R}^d. \tag{7}$$

A differentiable function $f$ is said to be $L$-smooth if the gradient $\nabla f$ is $L$-Lipschitz continuous.

## 2 Stochastic Homotopy CGM

Most stochastic CGM variants require mini-batch size to increase, in order to reduce the variance of the gradient estimator. However, Mokhtari et al., [31] have recently shown that the following (biased) estimator (that can be implemented with a single sample) can be incorporated with the CGM analysis:

$$d_k = (1 - \rho_k)d_{k-1} + \rho_k \nabla_x f(x_k, \omega_k) \tag{8}$$

The resulting method guarantees $\mathcal{O}(1/k^{\frac{1}{3}})$ convergence rate for convex smooth minimization, but it does not apply to our composite problem template (P).

---

**Algorithm 1** SHCGM

**Input:** $x_1 \in \mathcal{X}$, $\beta_0 > 0$, $d_0 = 0$
**for** $k = 1, 2, \ldots,$ **do**
    $\eta_k = 9/(k+8)$
    $\beta_k = \beta_0/(k+8)^{\frac{1}{2}}$
    $\rho_k = 4/(k+7)^{\frac{2}{3}}$
    $d_k = (1 - \rho_k)d_{k-1} + \rho_k \nabla_x f(x_k, \omega_k)$
    $v_k = d_k + \beta_k^{-1} A^\top \left(Ax_k - \mathrm{prox}_{\beta_k g}(Ax_k)\right)$
    $s_k = \arg\min_{x \in \mathcal{X}} \langle v_k, x \rangle$
    $x_{k+1} = x_k + \eta_k(s_k - x_k)$
**end for**

---

On the other hand, we introduced a CGM variant for composite problems (also covers affine constraints) in the deterministic setting in our prior work [41]. Our framework combines Nesterov smoothing [32] (and the quadratic penalty for affine constraints) with the CGM analysis. Unfortunately, this method does not work for stochastic problems.

In this paper, we propose the Stochastic Homotopy Conditional Gradient Method (SHCGM) for solving (P). The proposed method combines the stochastic CGM of [31] with our (deterministic) CGM for composite problems [41] in a non-trivial way.

Remark that the following formulation uniformly covers the Nesterov smoothing (with the Euclidean prox-function $\frac{1}{2}\|\cdot\|^2$) and the quadratic penalty (but the analyses for these two cases differ):

$$g_\beta(z) = \max_{y \in \mathbb{R}^d} \langle z, y \rangle - g^*(y) - \frac{\beta}{2}\|y\|^2, \quad \text{where} \quad g^*(x) = \max_{v \in \mathbb{R}^d} \langle x, v \rangle - g(v). \tag{9}$$

We call $g_\beta$ as the smooth approximation of $g$, parametrized by the penalty (or smoothing) parameter $\beta > 0$. It is easy to show that $g_\beta$ is $1/\beta$-smooth. Remark that the gradient of $g_\beta$ can be computed by the following formula:

$$\nabla_x g_\beta(Ax) = A^\top \mathrm{prox}_{\beta^{-1}g^*}(\beta^{-1}Ax) = \beta^{-1} A^\top \left(Ax - \mathrm{prox}_{\beta g}(Ax)\right), \tag{10}$$

where the second equality follows from the Moreau decomposition.

The main idea is to replace the non-smooth component $g$ by the smooth approximation $g_\beta$ in (P). Clearly the solutions for (P) with $g(Ax)$ and $g_\beta(Ax)$ do not coincide for any value of $\beta$. However, $g_\beta \to g$ as $\beta \to 0$. Hence, we adopt a homotopy technique: We decrease $\beta$ at a controlled rate as we progress in the optimization procedure, so that the decision variable converges to a solution of the original problem.

SHCGM is characterized by the following iterative steps:
▷ Decrease the step-size, smoothing and gradient averaging parameters $\eta_k$, $\beta_k$ and $\rho_k$.
▷ Call the stochastic first-order oracle and compute the gradient estimator $d_k$ in (8).
▷ Compute the gradient estimator $v_k$ for the smooth approximation of the composite objective,

$$F_{\beta_k}(x) = \mathbb{E}_\Omega f(x, \omega) + g_{\beta_k}(Ax) \quad \implies \quad v_k = d_k + \nabla_x g_{\beta_k}(Ax). \qquad (11)$$

▷ Compute the *lmo* with respect to $v_k$.
▷ Perform a CGM step to find the next iterate.

The roles of $\rho_k$ and $\beta_k$ are coupled. The former controls the variance of the gradient estimator, and the latter decides how fast we reduce the smoothing parameter to approach to the original problem. A carefully tuned interaction between these two parameters allows us to prove the following convergence rates.

**Assumption (Bounded variance).** We assume the following bounded variance condition holds:

$$\mathbb{E}\left[\|\nabla_x f(x, \omega) - \nabla_x \mathbb{E}_\Omega f(x, \omega)\|^2\right] \leq \sigma^2 < +\infty. \qquad (12)$$

**Theorem 1** (Lipschitz-continuous regularizer). *Assume that $g : \mathbb{R}^d \to \mathbb{R}$ is $L_g$-Lipschitz continuous. Then, the sequence $x_k$ generated by Algorithm 1 satisfies the following convergence bound:*

$$\mathbb{E}F(x_{k+1}) - F^\star \leq 9^{\frac{1}{3}} \frac{C}{(k+8)^{\frac{1}{3}}} + \frac{\beta_0 L_g^2}{2\sqrt{k+8}}, \qquad (13)$$

*where $C := \frac{81}{2} D_\mathcal{X}^2 (L_f + \beta_0 \|A\|^2) + 36\sigma D_\mathcal{X} + 27\sqrt{3} L_f D_\mathcal{X}^2$.*

*Proof sketch.* The proof follows the following steps:
(i) Relate the stochastic gradient to the full gradient (Lemma 7).
(ii) Show convergence of the gradient estimator to full gradient (Lemma 8).
(iii) Show $\mathcal{O}(1/k^{\frac{1}{3}})$ convergence rate on the smooth gap $\mathbb{E}F_{\beta_k}(x_{k+1}) - F^\star$ (Theorem 9).
(iv) Translate this bound to the actual sub-optimality $\mathbb{E}F(x_{k+1}) - F^\star$ by using the envelope property for Nesterov smoothing, see Equation (2.7) in [32]. □

Convergence rate guarantees for stochastic CGM with Lipschitz continuous $g$ (also based on Nesterov smoothing) are already known in the literature, see [16, 22, 23] for examples. Our rate is not faster than the ones in [22, 23], but we obtain $\mathcal{O}(\frac{1}{\epsilon^3})$ sample complexity in the statistical setting as opposed to $\mathcal{O}(\frac{1}{\epsilon^4})$.

In contrast with the existing stochastic CGM variants, our algorithm can also handle affine constraints. Remark that the indicator functions are not Lipschitz continuous, hence the Nesterov smoothing technique does not work for affine constraints.

**Assumption (Strong duality).** For problems with affine constraints, we further assume that the strong duality holds. Slater's condition is a common sufficient condition for strong duality. By Slater's condition, we mean

$$\text{relint}(\mathcal{X} \times \mathcal{K}) \cap \{(x, r) \in \mathbb{R}^n \times \mathbb{R}^d : Ax = r\} \neq \emptyset. \qquad (14)$$

Recall that the strong duality ensures the existence of a finite dual solution.

**Theorem 2** (Affine constraints). *Suppose that $g : \mathbb{R}^d \to \mathbb{R}$ is the indicator function of a simple convex set $\mathcal{K}$. Assuming that the strong duality holds, the sequence $x_k$ generated by SHCGM satisfies*

$$\mathbb{E}\mathbb{E}_\Omega f(x_{k+1}, \omega) - f^\star \geq -\|y^\star\| \, \mathbb{E}\text{dist}(Ax_{k+1}, \mathcal{K})$$

$$\mathbb{E}\mathbb{E}_\Omega f(x_{k+1}, \omega) - f^\star \leq 9^{\frac{1}{3}} \frac{C}{(k+8)^{\frac{1}{3}}} \qquad (15)$$

$$\mathbb{E}\text{dist}(Ax_{k+1}, \mathcal{K}) \leq \frac{2\beta_0 \|y^\star\|}{\sqrt{k+8}} + \frac{2\sqrt{2 \cdot 9^{\frac{1}{3}} C \beta_0}}{(k+8)^{\frac{5}{12}}}$$

*Proof sketch.* We re-use the ingredients of the proof of Theorem 1, except that at step (iv) we translate the bound on the smooth gap (penalized objective) to the actual convergence measures (objective residual and feasibility gap) by using the Lagrange saddle point formulations and the strong duality. See Corollaries 1 and 2. $\qquad\square$

**Remark (Comparison to baseline).** SHCGM combines ideas from [31] and [41]. Surprisingly,

$\triangleright$ $\mathcal{O}(1/k^{\frac{1}{3}})$ rate in objective residual matches the rate in [31] for smooth minimization.

$\triangleright$ $\mathcal{O}(1/k^{\frac{5}{12}})$ rate in feasibility gap is only an order of $k^{\frac{1}{12}}$ worse than the deterministic variant in [41].

**Remark (Inexact oracles).** We assume to use the exact solutions of *lmo* in SHCGM in Theorems 1 and 2. In many applications, however, it is much easier to find an approximate solution of *lmo*. For instance, this is the case for the SDP problems in Section 1.1. To this end, we extend our results for inexact *lmo* calls with additive and multiplicative error in the supplements.

**Remark (Splitting).** An important use-case of affine constraints in (P) is splitting (see Section 5.6 in [41]). Suppose that $\mathcal{X}$ can be written as the intersection of two (or more) simpler (in terms of computational cost of *lmo* or projection) sets $\mathcal{A} \cap \mathcal{B}$. By using the standard product space technique, we can reformulate this problem in the extended space $(x, y) \in \mathcal{A} \times \mathcal{B}$ with the constraint $x = y$:

$$\underset{(x,y)\in\mathcal{A}\times\mathcal{B}}{\text{minimize}} \quad \mathbb{E}_\Omega f(x,\omega) \quad \text{subject to} \quad x = y. \tag{16}$$

This allows us to decompose the difficult optimization domain $\mathcal{X}$ into simpler pieces. SHCGM requires *lmo* of $\mathcal{A}$ and *lmo* $\mathcal{B}$ separately. Alternatively, we can also use the projection onto one of the component sets (say $\mathcal{B}$) by reformulating the problem in domain $\mathcal{A}$ with an affine constraint $x \in \mathcal{B}$:

$$\underset{x\in\mathcal{A}}{\text{minimize}} \quad \mathbb{E}_\Omega f(x,\omega) \quad \text{subject to} \quad x \in \mathcal{B}. \tag{17}$$

An important example is the completely positive cone (intersection of the positive-semidefinite cone and the first orthant). Remark that the Clustering SDP example in Section 1.1 is also defined on this cone. While the *lmo* of this intersection can only be evaluated in $\mathcal{O}(n^3)$ computetion by using the Hungarian method, we can compute the *lmo* for the semidefinite cone and the projection onto the first orthant much more efficiently.

## 3 Related Works

CGM dates back to the 1956 paper of Frank and Wolfe [8]. It did not acquire much interest in machine learning until the last decade because of its slower convergence rate in comparison with the (projected) accelerated gradient methods. However, there has been a resurgence of interest in CGM and its variants, following the seminal papers of Hazan [14] and Jaggi [18]. They demonstrate that CGM might offer superior computational complexity than state-of-the-art methods in many large-scale optimization problems (that arise in machine learning) despite its slower convergence rate, thanks to its lower per-iteration cost.

The original method by Frank and Wolfe [8] was proposed for smooth convex minimization on polytopes. The analysis is extended for smooth convex minimization on simplex by Clarkson [3], spactrahedron by Hazan [14], and finally for arbitrary compact convex sets by Jaggi [18]. All these methods are restricted for smooth problems.

Lan [21] proposed a variant for non-smooth minimization based on the Nesterov smoothing technique. Lan and Zhou [23] also introduced the conditional gradient sliding method and extended it for the non-smooth minimization in a similar way. These methods, however, are not suitable for solving (P) because we let $g$ to be an indicator function which is not smoothing friendly.

In a prior work [41], we introduced homotopy CGM (HCGM) for composite problems (also with affine constraints). HCGM combines the Nesterov smoothing and quadratic penalty techniques under the CGM framework. It has $\mathcal{O}(1/\varepsilon^2)$ iteration complexity. In a follow-up work [40], we extended this method from quadratic penalty to an augmented Lagrangian formulation for empirical benefits. Gidel et al., [10] also proposed an augmented Lagrangian CGM but the analysis and guarantees differ. We refer to the references in [40, 41] for other variants in this direction.

So far, we have focused on deterministic variants of CGM. The literature on stochastic variants are much younger. We can trace it back to the Hazan and Kale's projection-free methods for online

learning [16]. When $g$ is a non-smooth but Lipschitz continuous function, their method returns an $\varepsilon$-solution in $\mathcal{O}(1/\varepsilon^4)$ iterations.

The standard extension of CGM to the stochastic setting gets $\mathcal{O}(1/\varepsilon)$ iteration complexity for smooth minimization, but with an increasing minibatch size. Overall, this method requires $\mathcal{O}(1/\varepsilon^3)$ sample complexity, see [17] for the details. More recently, Mokhtari et al., [31] proposed a new variant with $\mathcal{O}(1/\varepsilon^3)$ convergence rate, but the proposed method can work with a single sample at each iteration. Hazan and Luo [17] and Yurtsever et al., [42] incorporated various variance for further improvements. Goldfarb et al., [11] introduced two stochastic CGM variants, with away-steps and pairwise-steps. These methods enjoy linear convergence rate (however, the batchsize increases exponentially) but for strongly convex objectives and only in polytope domains. None of these stochastic CGM variants work for non-smooth (or composite) problems.

Non-smooth conditional gradient sliding by Lan and Zhou [23] also have extensions to the stochastic setting. There is also a lazy variant with further improvements by Lan et al., [22]. Note however, similar to their deterministic variants, these methods are based on the Nesterov smoothing and are not suitable for problems with affine constraints.

Garber and Kaplan [9] considers problem (P). They also propose a variance reduced algorithm, but this method indeed solves the smooth relaxation of (P) (see Definition 1 Section 4.1). Contrary to SHCGM, this method might not asymptotically converge to a solution of the original problem.

Lu and Freund [28] also studied a similar problem template. However, their method incorporates the non-smooth term into the linear minimization oracle. This is restrictive in practice because the non-smooth term can increase the cost of linear minimization. In particular, this is the case when $g$ is an indicator function, such as in SDP problems. This is orthogonal to our scenario in which the affine constraints are processed by smoothing, not directly through *lmo*.

In recent years, CGM has also been extended for non-convex problems. These extensions are beyond the scope of this paper. We refer to Yu et al., [39] and Julien-Lacoste [19] for the non-convex extensions in the deterministic setting, and to Reddi et al., [34], Yurtsever et al., [42], and Shen et al. [37] in the stochastic setting.

To the best of our knowledge, SHCGM is the first CGM-type algorithm for solving (P) with cheap linear minimization oracles. Another popular approach for solving large-scale instances of (P) is the operator splitting. See [2] and the references therein for stochastic operator splitting methods. Unfortunately, these methods still require projection onto $\mathcal{X}$ at each iteration. This projection is arguably more expensive than the linear minimization. For instance, for solving (3), the projection has cubic cost (with respect to the problem dimension $n$) while the linear minimization can be efficiently solved using subspace iterations, as depicted in Table 1.

| Algorithm | Iteration complexity | Sample complexity | Solves (3) | Per-iteration cost (for (3)) |
|---|---|---|---|---|
| [41] | $\mathcal{O}(1/\varepsilon^2)$ | $N$ | Yes | $\Theta(N_\nabla/\delta)$ |
| [9] | $\mathcal{O}(1/\varepsilon^2)$ | $\mathcal{O}(1/\varepsilon^4)$ | No | $\Theta(N_\nabla/\delta)$ |
| [17] | $\mathcal{O}(1/\varepsilon)$ | $\mathcal{O}(1/\varepsilon^3)$ | No | $\Theta(N_\nabla/\delta)$ |
| [28] | $\mathcal{O}(1/\varepsilon)$ | $\mathcal{O}(1/\varepsilon^2)$ | No | SDP |
| [15] | $\mathcal{O}(1/\varepsilon)$ | $N$ | No | $\Theta(N_\nabla/\delta)$ |
| [2]* | $-$ | $-$ | Yes | $\Theta(n^3)$ |
| **SHCGM** | $\mathcal{O}(1/\varepsilon^3)$ | $\mathcal{O}(1/\varepsilon^3)$ | Yes | $\Theta(N_\nabla/\delta)$ |

Table 1: Existing algorithms to tackle (3). $N$ is the size of the dataset. $n$ is the dimension of each datapoint. $N_\nabla$ is the number of non-zeros of the gradient. $\delta$ is the accuracy of the approximate *lmo*. The per-iteration cost of [28] is the cost of solving a SDP in the canonical form.
*[2] has $\mathcal{O}(1/\varepsilon^2)$ iteration and sample complexity when the objective function is strongly convex. This is not the case in our model problem, and [2] only has an asymptotic convergence guarantee.

# 4 Numerical Evidence

This section presents the empirical performance of the proposed method for the stochastic k-means clustering, covariance matrix estimation, and matrix completion problems. We performed the experiments in MATLAB R2018a using a computing system of $4\times$ Intel Xeon CPU E5-2630 v3@2.40GHz and 16 GB RAM. We include the code to reproduce the results in the supplements.

## 4.1 Stochastic k-means Clustering

We consider the SDP formulation (4) of the k-means clustering problem. The same problem is used in numerical experiments by Mixon et al. [30], and we design our experiment based on their problem setup[1] with a sample of $1000$ datapoints from the MNIST data[2]. See [30] for details on the preprocessing.

We solve this problem with SHCGM and compare it against HCGM [41] as the baseline. HCGM is a deterministic algorithm hence it uses the full gradient. For SHCGM, we compute a gradient estimator by randomly sampling 100 datapoints at each iteration. Remark that this corresponds to observing approximately 1 percent of the entries of $D$.

We use $\beta_0 = 1$ for HCGM and $\beta_0 = 10$ for SHCGM. We set these values by tuning both methods by trying $\beta_0 = 0.01, 0.1, ..., 1000$. We display the results in Figure 1 where we denote a full pass over the entries of $D$ as an epoch. Figure 1 demonstrates that SHCGM performs similar to HCGM although it uses less data.

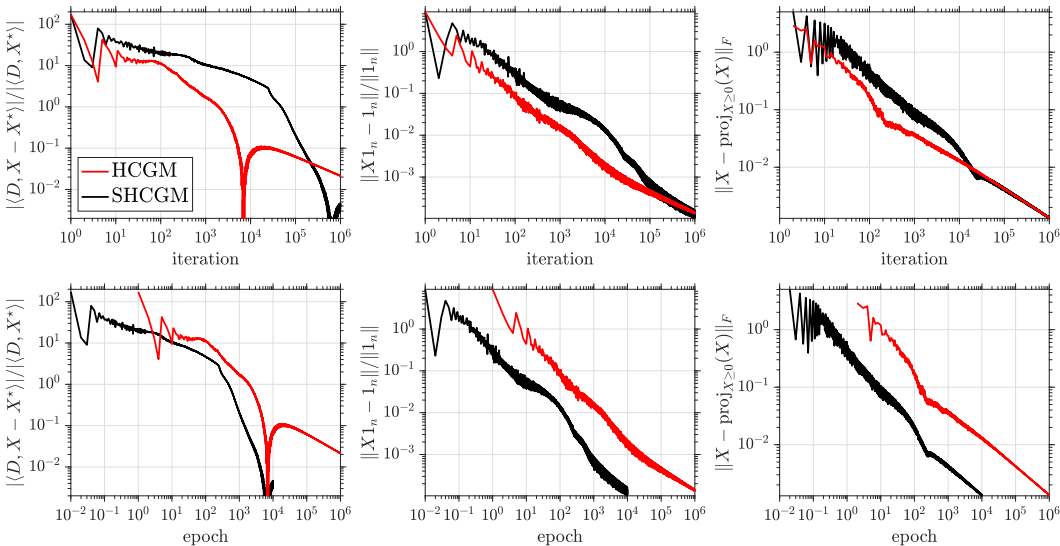

Figure 1: Comparison of SHCGM with HCGM for k-means clustering SDP in Section 4.1.

## 4.2 Online Covariance Matrix Estimation

Covariance matrix estimation is an important problem in multivariate statistics with applications in many fields including gene microarrays, social network, finance, climate analysis [35, 36, 7, 6], etc. In the online setting, we suppose that the data is received as a stream of datapoints in time.

The deterministic approach is to first collect some data, and then to train an empirical risk minimization model using the data collected. This has obvious limitations, since it may not be clear a priori how much data is enough to precisely estimate the covariance matrix. Furthermore, data can be too large to store or work with as a batch. To this end, we consider an online learning setting. In this case, we use each datapoint as it arrives and then discard it.

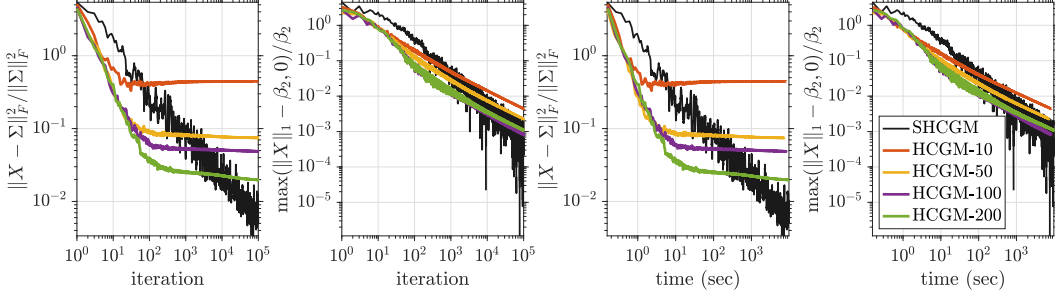

Figure 2: SHCGM and HCGM on Online covariance matrix estimation from streaming data.

Let us consider the following sparse covariance matrix estimation template (this template also covers other problems such as graph denoising and link prediction [35]) :

$$\underset{X \in \mathbb{S}_+^n, \ \mathrm{tr}(X) \leq \beta_1}{\text{minimize}} \quad \mathbb{E}_\Omega \|X - \omega\omega^\top\|_F^2 \quad \text{subject to} \quad \|X\|_1 \leq \beta_2. \tag{18}$$

where $\|X\|_1$ denotes the $\ell_1$ norm (sum of absolute values of the entries).

Our test setup is as follows: We first create a block diagonal covariance matrix $\Sigma \in \mathbb{R}^{n \times n}$ using 10 blocks of the form $\phi\phi^\top$, where entries of $\phi$ are drawn uniformly random from $[-1, 1]$. This gives us a sparse matrix $\Sigma$ of rank 10. Then, as for datapoints, we stream observations of $\Sigma$ in the form $\omega_i \sim \mathcal{N}(0, \Sigma)$. We fix the problem dimension $n = 1000$.

We compare SHCGM with the deterministic method, HCGM. We use $\beta_0 = 1$ for both methods. Both methods require the *lmo* for the positive-semidefinite cone with trace constraint, and the projection oracle for the $\ell_1$ norm constraint at each iteration.

We study two different setups: In Figure 2, we use SHCGM in the online setting. We sample a new datapoint at each iteration. HCGM, on the other hand, does not work in the online setting. Hence, we use the same sample of datapoints for all iterations. We consider 4 different cases with different sample sizes for HCGM, with 10, 50, 100 and 200 datapoints. Although this approach converges fast up to some accuracy, the objective value gets saturated at some estimation accuracy. Naturally, HCGM can achieve higher accuracy as the sample size increases.

We can also read the empirical convergence rates of SHCGM from Figure 2 as approximately $\mathcal{O}(k^{-1/2})$ for the objective residual and $\mathcal{O}(k^{-1})$ for the feasibility gap, significantly better than the theoretical guarantees .

If we can store larger samples, we can also consider minibatches for the stochastic methods. Figure 3 compares the deterministic approach with 200 datapoints with the stochastic approach with minibatch size of 200. In other words, while the deterministic method uses the same 200 datapoints for all iterations, we use a new draw of 200 datapoints at each iteration with SHCGM.

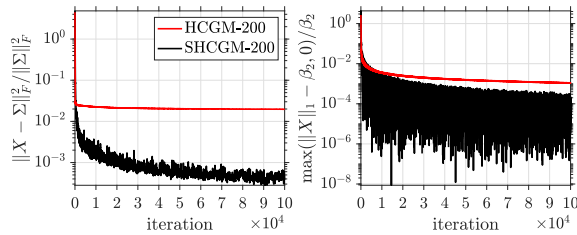

Figure 3: Comparison of SHCGM with HCGM batchsize 200 for online covariance matrix estimation.

### 4.3 Stochastic Matrix Completion

We consider the problem of matrix completion with the following mathematical formulation:

$$\underset{\|X\|_* \leq \beta_1}{\text{minimize}} \quad \sum_{(i,j) \in \Omega} (X_{i,j} - Y_{i,j})^2 \quad \text{subject to} \quad 1 \leq X \leq 5, \tag{19}$$

where, $\Omega$ is the set of observed ratings (samples of entries from the true matrix $Y$ that we try to recover), and $\|X\|_*$ denotes the nuclear-norm (sum of singular values). The affine constraint $1 \leq X \leq 5$ imposes a hard threshold on the estimated ratings (in other words, the entries of $X$).

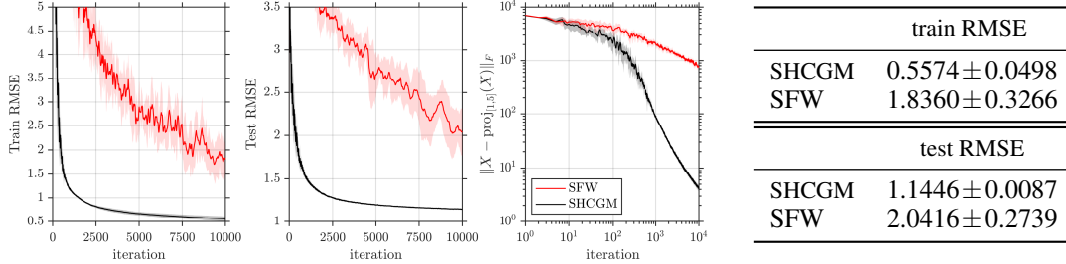

| | train RMSE | |
|---|---|---|
| SHCGM | $0.5574 \pm 0.0498$ | |
| SFW | $1.8360 \pm 0.3266$ | |
| | test RMSE | |
| SHCGM | $1.1446 \pm 0.0087$ | |
| SFW | $2.0416 \pm 0.2739$ | |

Figure 4: Training Error, Feasibility gap and Test Error for MovieLens 100k. Table shows the mean values and standard deviation of train and test RMSE over 5 different train/test splits at the end of $10^4$ iterations.

We first compare SHCGM with the Stochastic Frank-Wolfe (SFW) from [31]. We consider a test setup with the MovieLens100k dataset[3] [13]. This dataset contains $\sim 100$'000 integer valued ratings between 1 and 5, assigned by 1682 users to 943 movies. The aim of this experiment is to emphasize the flexibility of SHCGM: Recall that SFW does not directly apply to (19) as it cannot handle the affine constraint $1 \leq X \leq 5$. Therefore, we apply SFW to a relaxation of (19) that omits this constraint. Then, we solve (19) with SHCGM and compare the results.

We use the default `ub.train` and `ub.test` partitions provided with the original data. We set the model parameter for the nuclear norm constraint $\beta_1 = 7000$, and the initial smoothing parameter $\beta_0 = 10$. At each iteration, we compute a gradient estimator from 1000 *iid* samples. We perform the same test independently for 10 times to compute the average performance and confidence intervals. In Figure 4, we report the training and test errors (root mean squared error) as well as the feasibility gap. The solid lines display the average performance, and the shaded areas show $\pm$ one standard deviation. Note that SHCGM performs uniformly better than SFW, both in terms of the training and test errors. The Table shows the values achieved at the end of $10'000$ iterations.

Finally, we compare SHCGM with the stochastic three-composite convex minimization method (S3CCM) from [43]. S3CCM is a projection-based method that applies to (19). In this experiment, we aim to demonstrate the advantages of the projection-free methods for problems in large-scale.

We consider a test setup with the MovieLens1m dataset[3] with $\sim 1$ million ratings from $\sim 6000$ users on $\sim 4000$ movies. We partition the data into training and test samples with a $80/20$ train/test split. We use $10'000$ *iid* samples at each iteration to compute a gradient estimator. We set the model parameter $\beta_1 = 20'000$. We use $\beta_0 = 10$ for SHCGM, and we set the step-size parameter $\gamma = 1$ for S3CCM. We implement the *lmo* efficiently using the power method. We refer to the code in the supplements for details on the implementation.

Figure 5 reports the outcomes of this experiment. SHCGM clearly outperforms S3CCM in this test. We run both methods for 2 hours. Within this time limit, SHCGM can perform $27'860$ iterations while S3CCM can gets only up to 435 because of the high computational cost of the projection.

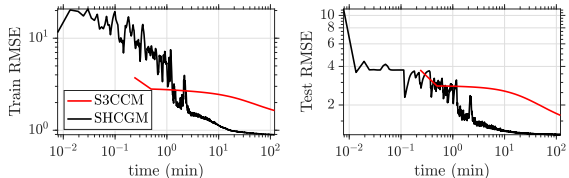

Figure 5: SHCGM vs S3CCM with MovieLens-1M.

## 5   Conclusions

We introduced a scalable stochastic CGM-type method for solving convex optimization problems with affine constraints and demonstrated empirical superiority of our approach in various numerical experiments. In particular, we consider the case of stochastic optimization of SDPs for which we give the first projection-free algorithm. In general, we showed that our algorithm provably converges to an optimal solution of (P) with $\mathcal{O}(k^{-1/3})$ and $\mathcal{O}(k^{-5/12})$ rates in the objective residual and feasibility gap respectively, with a sample complexity in the statistical setting of $\mathcal{O}(k^{-1/3})$. The possibility of a faster rate with the same (or even better) sample complexity remains an open question as well as an adaptive approach with $\mathcal{O}(k^{-1/2})$ rate when fed with exact gradients.

## Acknowledgements

Francesco Locatello has received funding from the Max Planck ETH Center for Learning Systems, by an ETH Core Grant (to Gunnar Rätsch) and by a Google Ph.D. Fellowship. Volkan Cevher and Alp Yurtsever have received funding from the Swiss National Science Foundation (SNSF) under grant number $200021\_178865/1$, and the European Research Council (ERC) under the European Union's Horizon 2020 research and innovation program (grant agreement no 725594 - time-data).

## Footnotes

[1]D.G. Mixon, S. Villar, R.Ward. — Available at `https://github.com/solevillar/kmeans_sdp`

[2]Y. LeCun and C. Cortes. — Available at `http://yann.lecun.com/exdb/mnist/`

[3]F.M. Harper, J.A. Konstan. — Available at `https://grouplens.org/datasets/movielens/`

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
