[Supplementary Material]

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

# A  A Review of Smoothing

The technique described in [32] consists in a the following smooth approximation of a Lipschitz continuous function $g$ as:

$$g_\beta(z) = \max_{y \in \mathbb{R}^d} \langle z, y \rangle - g^*(y) - \frac{\beta}{2}\|y\|^2,$$

where $\beta > 0$ controls the tightness of smoothing and $g^*$ denotes the Fenchel conjugate of $g$

$$g^*(x) = \sup_{v \in \text{dom } g} \langle x, v \rangle - g(v).$$

It is easy to see that $g_\beta$ is convex and $\frac{1}{\beta}$ smooth. Optimizing $g_\beta(z)$ guarantees progress on $g(z)$ when $g(z)$ is $L_g$-Lipschitz continuous as:

$$g_\beta(z) \leq g(z) \leq g_\beta(z) + \frac{\beta}{2}L_g$$

The challenge of smoothing an affine constraint consists in the fact that the indicator function is not Lipschitz. Therefore, $g^*$ does not have bounded support so adding a strongly convex term to it does not guarantee that $g$ and its smoothed version are uniformly close.

In order to smooth constraints which are not Nesterov smoothable, [41] consider an Homotopy transformation on $\beta$ which can be intuitively understood as follows. If $\beta$ decreases during the optimization, optimizing $g_\beta(z)$ will progressively become similar to optimizing $g(z)$. Therefore, the iterate will converge to the feasibility set.

Using the Homotopy smoothing, the objective of Equation (P) is replaced by the following approximation:

$$\min_{x \in \mathcal{X}} F_\beta(x) := \mathbb{E}_\Omega f(x, \omega) + g_\beta(Ax). \tag{20}$$

Let $y^*_{\beta_k}$ be:

$$y^*_{\beta_k}(Ax) = \arg\max_{y \in \mathbb{R}^d} \langle Ax, y \rangle - g^*(y) - \frac{\beta_k}{2}\|y\|^2 = \text{prox}_{\beta_k^{-1}g^*}(\beta_k^{-1}Ax) = \frac{1}{\beta_k}\big(Ax - \text{prox}_{\beta_k g}(Ax)\big),$$

with the last equality due to the Moreau decomposition.

Note that often $\text{prox}_{\beta_k g}(Ax)$ is easy to compute (for example when $g(z)$ is an affine constraint) but the projection on $\mathcal{X}$ is not. For example, for the $AX = b$ constraint of (3), $y^*_{\beta_k}(AX) = AX - b$. Therefore, [41] suggests to follow the same iterative procedure of the CGM which queries a Linear Minimization Oracle (lmo) at each iteration:

$$\text{lmo}(\nabla F_{\beta_k}) := \arg\min_{x \in \mathcal{X}} \langle x, \nabla F_{\beta_k}(x_k) \rangle \tag{21}$$

Moreover, we can compute the gradient of $F_{\beta_k}$ as long as $\text{prox}_{\beta_k g}(Ax)$ is easy to compute. Indeed:

$$\nabla F_{\beta_k}(x) = \nabla_x \mathbb{E}_\Omega f(x, \omega) + A^\top y^*_{\beta_k}(Ax). \tag{22}$$

The solution of the *lmo* is then combined to the current iterate with a convex combination, so that the next iterate is guaranteed to be a member of $\mathcal{X}$. In the deterministic setting this technique comes with a reduction in the rate from $\mathcal{O}(1/k)$ to $\mathcal{O}(1/\sqrt{k})$.

# B  Inexact Oracles

In practice, finding an exact solution can be expensive, especially when it involves a matrix factorization. Therefore, algorithms which are robust against inexact oracles are crucial in practice.

Even when the penalty is not present, we are not aware of approximate oracle rates in the framework of [31]. Due to the accumulation of the stochastic gradient, the deterministic definitions of inexact oracle are applicable to the stochastic case with the non-smooth penalty [20, 25, 27].

## B.1 Additive Error

At iteration $k$, for the given $v_k$, we assume that the approximate *lmo* returns an element $\tilde{s}_k \in \mathcal{X}$ such that:

$$\langle v_k,\, \tilde{s}_k \rangle \leq \langle v_k,\, s_k \rangle + \delta \frac{\eta_k}{2} D_{\mathcal{X}}^2 \left( L_f + \frac{\|A\|^2}{\beta_k} \right) \tag{23}$$

for some $\delta > 0$, where $s_k$ is the exact *lmo* solution.

We now present the convergence guarantees of Algorithm 1 when the exact *lmo* is replaced with the approximate oracle with additive error.

**Corollary 3.** *Assume that $g$ is $L_g$-Lipschitz continuous. Then, the sequence $x_k$ generated by Algorithm 1 for $k \geq 1$ with approximate lmo* (23) *satisfies:*

$$\mathbb{E}F(x_{k+1}) - F^\star \leq 9^{\frac{1}{3}} \frac{C_\delta}{(k+8)^{\frac{1}{3}}} + \frac{\beta_0 L_g^2}{2\sqrt{k+8}},$$

*where $C_\delta := \frac{81}{2} D_{\mathcal{X}}^2 (L_f + \beta_0 \|A\|^2)(1+\delta) + 9 D_{\mathcal{X}} \sqrt{Q}$. We can optimize $\beta_0$ from this bound if $\delta$ is known.*

**Corollary 4.** *Assume that $g$ is the indicator function of a simple convex set $\mathcal{K}$. Then, the sequence $x_k$ generated by Algorithm 1 with the lmo* (23) *satisfies:*

$$\mathbb{E}\mathbb{E}_\Omega f(x_{k+1}, \omega) - f^\star \geq -\|y^\star\| \, \mathbb{E}\mathrm{dist}(Ax_{k+1}, \mathcal{K})$$

$$\mathbb{E}\mathbb{E}_\Omega f(x_{k+1}, \omega) - f^\star \leq 9^{\frac{1}{3}} \frac{C_\delta}{(k+8)^{\frac{1}{3}}}$$

$$\mathbb{E}\mathrm{dist}(Ax_{k+1}, \mathcal{K}) \leq \frac{2\beta_0 \|y^\star\|}{\sqrt{k+8}} + \frac{2\sqrt{2 \cdot 9^{\frac{1}{3}} C_\delta \beta_0}}{(k+8)^{\frac{5}{12}}}$$

## B.2 Multiplicative Error

The additive error requires the accuracy of *lmo* to increase as the algorithm progresses [18]. This is restrictive in practice as it forces to invest more and more effort in the solution of the *lmo* problem.

For this reason, multiplicative error is often preferred, even though it adds the quality of the *lmo* as a hyperparmeter [25, 26], which we consider:

$$\langle v_k,\, \tilde{s}_k - x_k \rangle \leq \delta \langle v_k,\, s_k - x_k \rangle \tag{24}$$

where $\delta \in (0, 1]$ and $s_k$ is the exact *lmo* solution.

We now present the convergence guarantees of Algorithm 1 when the exact *lmo* is replaced with the approximate oracle with multiplicative error (24)

**Corollary 5.** *Assume that $g$ is $L_g$-Lipschitz continuous. Then, the sequence $x_k$ generated by Algorithm 1 with the lmo* (24)*, and modifying $\eta_k = \frac{9}{\delta(k-1)+9}$, $\beta_k = \frac{\beta_0}{\sqrt{\delta(k-1)+9}}$ and $\rho_k = \frac{4}{(\delta(k-2)+9)^{\frac{2}{3}}}$ satisfies:*

$$\mathbb{E}F(x_{k+1}) - F^\star \leq 9^{\frac{1}{3}} \frac{\frac{C}{\delta} + \mathcal{E}_1}{(\delta(k-1)+9)^{\frac{1}{3}}} + \frac{\beta_0 L_g^2}{2\sqrt{\delta(k-1)+9}},$$

*We can optimize $\beta_0$ from this bound if $\delta$ is known.*

**Corollary 6.** *Assume that $g$ is the indicator function of a simple convex set $\mathcal{K}$. Then, the sequence $x_k$ generated by Algorithm 1 with approximate lmo* (24)*, and modifying $\eta_k = \frac{9}{\delta(k-1)+9}$, $\beta_k = $*

$\frac{\beta_0}{\sqrt{\delta(k-1)+9}}$ and $\rho_k = \frac{4}{(\delta(k-2)+9)^{\frac{2}{3}}}$ satisfies:

$$\mathbb{E}\mathbb{E}_\Omega f(x_{k+1}, \omega) - f^\star \geq -\|y^\star\| \, \mathbb{E}\mathrm{dist}(Ax_{k+1}, \mathcal{K})$$

$$\mathbb{E}\mathbb{E}_\Omega f(x_{k+1}, \omega) - f^\star \leq 9^{\frac{1}{3}} \frac{\frac{C}{\delta} + \mathcal{E}_1}{(\delta(k-1)+9)^{\frac{1}{3}}}$$

$$\mathbb{E}\mathrm{dist}(Ax_{k+1}, \mathcal{K}) \leq \frac{2\beta_0\|y^\star\|}{\sqrt{\delta(k-1)+9}} + \frac{2\sqrt{2 \cdot 9^{\frac{1}{3}}(\frac{C}{\delta} + \mathcal{E}_1)\beta_0}}{(\delta(k-1)+9)^{\frac{5}{12}}}$$

## C  Convergence Rate

We first prove some key lemmas. This section builds on top of the analysis of [31] and the homotopy CGM framework. All these results are for the inexact oracle with additive error, the exact oracle case can be obtained setting $\delta = 0$.

**Lemma 7.** *For any given iteration $k \geq 1$ of Algorithm 1 the following relation holds:*

$$\langle \nabla F_{\beta_k}(x_k), \, s_k - x_k \rangle \leq \|\nabla_x \mathbb{E}_\Omega f(x_k, \omega) - d_k\| D_{\mathcal{X}} + f^\star - \mathbb{E}_\Omega f(x_k, \omega) + g(Ax^\star) - g_{\beta_k}(Ax_k)$$
$$- \frac{\beta_k}{2}\|y^*_{\beta_k}(Ax_k)\|^2 + \delta\frac{\eta_k}{2}D_{\mathcal{X}}^2\left(L_f + \frac{\|A\|^2}{\beta_k}\right)$$

*where $\delta \geq 0$ is the accuracy of the inexact lmo with additive error.*

*Proof.*

$$\langle \nabla F_{\beta_k}(x_k), \, \tilde{s}_k - x_k \rangle = \langle \nabla_x \mathbb{E}_\Omega f(x_k, \omega), \, \tilde{s}_k - x_k \rangle + \langle A^\top \nabla g_{\beta_k}(Ax_k), \, \tilde{s}_k - x_k \rangle$$
$$= \langle \nabla_x \mathbb{E}_\Omega f(x_k, \omega), \, \tilde{s}_k - x_k \rangle + \langle A^\top \nabla g_{\beta_k}(Ax_k), \, \tilde{s}_k - x_k \rangle$$
$$\quad + \langle d_k, \, \tilde{s}_k - x_k \rangle - \langle d_k, \, \tilde{s}_k - x_k \rangle$$
$$= \langle \nabla_x \mathbb{E}_\Omega f(x_k, \omega) - d_k, \, \tilde{s}_k - x_k \rangle + \langle d_k + A^\top \nabla g_{\beta_k}(Ax_k), \, \tilde{s}_k - x_k \rangle$$
$$\leq \langle \nabla_x \mathbb{E}_\Omega f(x_k, \omega) - d_k, \, s_k - x_k \rangle$$
$$\quad + \langle d_k + A^\top \nabla g_{\beta_k}(Ax_k), \, s_k - x_k \rangle + \delta\frac{\eta_k}{2}D_{\mathcal{X}}^2\left(L_f + \frac{\|A\|^2}{\beta_k}\right) \tag{25}$$
$$\leq \langle \nabla_x \mathbb{E}_\Omega f(x_k, \omega) - d_k, \, s_k - x_k \rangle$$
$$\quad + \langle d_k + A^\top \nabla g_{\beta_k}(Ax_k), \, x^\star - x_k \rangle + \delta\frac{\eta_k}{2}D_{\mathcal{X}}^2\left(L_f + \frac{\|A\|^2}{\beta_k}\right) \tag{26}$$
$$= \langle \nabla_x \mathbb{E}_\Omega f(x_k, \omega) - d_k, \, s_k - x_k \rangle + \langle d_k + A^\top \nabla g_{\beta_k}(Ax_k), \, x^\star - x_k \rangle$$
$$\quad + \langle \nabla_x \mathbb{E}_\Omega f(x_k, \omega), \, x^\star - x_k \rangle - \langle \nabla_x \mathbb{E}_\Omega f(x_k, \omega), \, x^\star - x_k \rangle + \delta\frac{\eta_k}{2}D_{\mathcal{X}}^2\left(L_f + \frac{\|A\|^2}{\beta_k}\right)$$
$$= \langle \nabla_x \mathbb{E}_\Omega f(x_k, \omega) - d_k, \, s_k - x^\star \rangle$$
$$\quad + \langle \nabla_x \mathbb{E}_\Omega f(x_k, \omega) + A^\top \nabla g_{\beta_k}(Ax_k), \, x^\star - x_k \rangle + \delta\frac{\eta_k}{2}D_{\mathcal{X}}^2\left(L_f + \frac{\|A\|^2}{\beta_k}\right)$$
$$\leq \|\nabla_x \mathbb{E}_\Omega f(x_k, \omega) - d_k\| \|s_k - x^\star\|$$
$$\quad + \langle \nabla_x \mathbb{E}_\Omega f(x_k, \omega) + A^\top \nabla g_{\beta_k}(Ax_k), \, x^\star - x_k \rangle + \delta\frac{\eta_k}{2}D_{\mathcal{X}}^2\left(L_f + \frac{\|A\|^2}{\beta_k}\right) \tag{27}$$
$$\leq \|\nabla_x \mathbb{E}_\Omega f(x_k, \omega) - d_k\| D_{\mathcal{X}}$$
$$\quad + \langle \nabla_x \mathbb{E}_\Omega f(x_k, \omega) + A^\top \nabla g_{\beta_k}(Ax_k), \, x^\star - x_k \rangle + \delta\frac{\eta_k}{2}D_{\mathcal{X}}^2\left(L_f + \frac{\|A\|^2}{\beta_k}\right) \tag{28}$$

where Equation (25) is the definition of inexact oracle with additive error, Equation (26) is because $s_k$ is a solution of $\min_{x\in\mathcal{X}} \langle d_k + A^\top \nabla g_{\beta_k}(Ax_k), \, x \rangle$, Equation (27) is Cauchy-Schwarz

and the Equation (28) is the definition of diameter. Now, convexity of $\mathbb{E}_\Omega f(x_k, \omega)$ ensures $\langle \nabla_x \mathbb{E}_\Omega f(x_k, \omega), x^\star - x_k \rangle \leq f^\star - \mathbb{E}_\Omega f(x_k, \omega)$. From Lemma 10 in [38] we have that:

$$g(z_1) \geq g_\beta(z_2) + \langle \nabla g_\beta(z_2),\, z_1 - z_2 \rangle + \frac{\beta}{2} \|y^*_\beta(z_2)\|^2. \tag{29}$$

Therefore:

$$\begin{aligned}
\langle A^\top \nabla g_{\beta_k}(Ax_k),\, x^\star - x_k \rangle &= \langle \nabla g_{\beta_k}(Ax_k),\, Ax^\star - Ax_k \rangle \\
&\leq g(Ax^\star) - g_{\beta_k}(Ax_k) - \frac{\beta_k}{2} \|y^*_{\beta_k}(Ax_k)\|^2.
\end{aligned}$$

Therefore:

$$\begin{aligned}
\langle \nabla F_{\beta_k}(x_k),\, \tilde{s}_k - x_k \rangle \leq{}& \|\nabla_x \mathbb{E}_\Omega f(x_k, \omega) - d_k\| D_\mathcal{X} + f^\star) - \mathbb{E}_\Omega f(x_k, \omega) + g(Ax^\star) - g_{\beta_k}(Ax_k) \\
&- \frac{\beta_k}{2} \|y^*_{\beta_k}(Ax_k)\|^2 + \delta \frac{\eta_k}{2} D_\mathcal{X}^2 \left( L_f + \frac{\|A\|^2}{\beta_k} \right)
\end{aligned}$$

$\square$

**Lemma 8.** *For any $k \geq 1$ the estimate of the gradient computed in Algorithm 1 satisfies:*

$$\mathbb{E}\left[ \|\nabla_x \mathbb{E}_\Omega f(x_k, \omega) - d_k\|^2 \right] \leq \frac{Q}{(k+8)^{\frac{2}{3}}}$$

*where $Q = \max\left\{ \|\nabla_x \mathbb{E}_\Omega f(x_1, \omega) - d_1\|^2 7^{\frac{2}{3}},\, 16\sigma^2 + 81 L_f^2 D_\mathcal{X}^2 \right\}$*

*Proof.* This lemma simply applies Lemma 1 and Lemma 17 of [31] to our different stepsizes. We report all the steps for clarity and completeness. First, we invoke Lemma 1:

$$\begin{aligned}
\mathbb{E}\left[ \|\nabla_x \mathbb{E}_\Omega f(x_k, \omega) - d_k\|^2 \right] &\leq \left(1 - \frac{\rho_k}{2}\right) \|\nabla_x \mathbb{E}_\Omega f(x_{k-1}, \omega) - d_{k-1}\|^2 + \rho_k^2 \sigma^2 + \frac{2 L_f^2 D_\mathcal{X}^2 \eta_{k-1}^2}{\rho_k} \\
&\leq \left(1 - \frac{2}{(k+7)^{\frac{2}{3}}}\right) \|\nabla_x \mathbb{E}_\Omega f(x_{k-1}, \omega) - d_{k-1}\|^2 + \frac{16\sigma^2 + 81 L_f^2 D_\mathcal{X}^2}{(k+7)^{\frac{4}{3}}} \tag{30}
\end{aligned}$$

where we used $\rho_k = \frac{4}{(k+7)^{\frac{2}{3}}}$. Now, Lemma 17 of [31] gives the following solution:

$$\phi_t \leq \frac{Q}{(k + k_0 + 1)^\alpha}$$

to the recursion

$$\phi_k \leq \left(1 - \frac{c}{(k+k_0)^\alpha}\right)\phi_{k-1} + \frac{b}{(k+k_0)^{2\alpha}}$$

where $b \geq 0$, $c > 1$, $\alpha \leq 1$, $k_0 \geq 0$ and $Q := \max\{\phi_1 k_0^\alpha, b/(c-1)\}$ Applying this lemma to Equation (30) with $k_0 = 7$, $\alpha = \frac{2}{3}$, $c = 2$, $b = 16\sigma^2 + 81 L_f^2 D_\mathcal{X}^2$ gives:

$$\mathbb{E}\left[ \|\nabla_x \mathbb{E}_\Omega f(x_k, \omega) - d_k\|^2 \right] \leq \frac{Q}{(k+8)^{\frac{2}{3}}}$$

where $Q = \max\left\{ \|\nabla \mathbb{E}_\Omega f(x_1, \omega) - d_1\|^2 7^{\frac{2}{3}},\, 16\sigma^2 + 81 L_f^2 D_\mathcal{X}^2 \right\}$ $\square$

### C.1 Proof of Theorem 9

We prove Theorem 9 with the oracle with additive error. The proof without additive error can be obtained with $\delta = 0$.

**Theorem 9.** *The sequence $x_k$ generated by Algorithm 1 satisfies the following bound for $k \geq 1$:*

$$\mathbb{E}F_{\beta_k}(x_{k+1}) - F^\star \leq 9^{\frac{1}{3}} \frac{C_\delta}{(k+8)^{\frac{1}{3}}},$$

*where $C_\delta := \frac{81}{2} D_\mathcal{X}^2 (L_f + \beta_0 \|A\|^2)(1 + \delta) + 9 D_\mathcal{X} \sqrt{Q}$, $Q = \max\left\{ 4\|\nabla \mathbb{E}_\Omega f(x_1, \omega) - d_1\|^2,\, 16\sigma^2 + 2 L_f^2 D_\mathcal{X}^2 \right\}$ and $\delta \geq 0$.*

*Proof.* Note that Theorem 9 can be obtained as a special case setting $\delta = 0$. First, we use the smoothness of $F_{\beta_k}$ to upper bound the progress. Note that $F_{\beta_k}$ is $(L_f + \|A\|^2/\beta_k)$-smooth.

$$F_{\beta_k}(x_{k+1}) \leq F_{\beta_k}(x_k) + \eta_k \langle \nabla F_{\beta_k}(x_k), \tilde{s}_k - x_k \rangle + \frac{\eta_k^2}{2} \|\tilde{s}_k - x_k\|^2 (L_f + \frac{\|A\|^2}{\beta_k})$$

$$\leq F_{\beta_k}(x_k) + \eta_k \langle \nabla F_{\beta_k}(x_k), \tilde{s}_k - x_k \rangle + \frac{\eta_k^2}{2} D_{\mathcal{X}}^2 (L_f + \frac{\|A\|^2}{\beta_k}), \tag{31}$$

where $s_k$ denotes the atom selected by the *lmo*, and the second inequality follows since $s_k \in \mathcal{X}$. We now apply Lemma 7 and obtain:

$$F_{\beta_k}(x_{k+1}) \leq F_{\beta_k}(x_k) + \eta_k \left( f^\star - \mathbb{E}_\Omega f(x_k, \omega) + g(Ax^\star) - g_{\beta_k}(Ax_k) - \frac{\beta_k}{2} \|\nabla y_{\beta_k}^\star(Ax_k)\|^2 \right)$$

$$+ \frac{\eta_k^2}{2} D_{\mathcal{X}}^2 (L_f + \frac{\|A\|^2}{\beta_k})(1 + \delta) + \eta_k \|\nabla_x \mathbb{E}_\Omega f(x_k, \omega) - d_k\| D_{\mathcal{X}} \tag{32}$$

$$= (1 - \eta_k) F_{\beta_k}(x_k) + \eta_k F^\star - \frac{\eta_k \beta_k}{2} \|\nabla y_{\beta_k}^\star(Ax_k)\|^2 + \frac{\eta_k^2}{2} D_{\mathcal{X}}^2 (L_f + \frac{\|A\|^2}{\beta_k})(1 + \delta)$$

$$+ \eta_k \|\nabla_x \mathbb{E}_\Omega f(x_k, \omega) - d_k\| D_{\mathcal{X}}.$$

Now, using Lemma 10 of [38] we get:

$$g_\beta(z_1) \leq g_\gamma(z_1) + \frac{\gamma - \beta}{2} \|y_\beta^\star(z_1)\|^2 \tag{33}$$

and therefore:

$$F_{\beta_k}(x_k) = \mathbb{E}_\Omega f(x_k, \omega) + g_{\beta_k}(Ax_k)$$

$$\leq \mathbb{E}_\Omega f(x_k, \omega) + g_{\beta_{k-1}}(Ax_k) + \frac{\beta_{k-1} - \beta_k}{2} \|y_{\beta_k}^\star(Ax_k)\|^2$$

$$= F_{\beta_{k-1}}(x_k) + \frac{\beta_{k-1} - \beta_k}{2} \|y_{\beta_k}^\star(Ax_k)\|^2.$$

We combine this with (32) and subtract $F^\star$ from both sides to get

$$F_{\beta_k}(x_{k+1}) - F^\star \leq (1 - \eta_k)\big(F_{\beta_{k-1}}(x_k) - F^\star\big) + \frac{\eta_k^2}{2} D_{\mathcal{X}}^2 (L_f + \frac{\|A\|^2}{\beta_k})(1 + \delta)$$

$$+ \big((1 - \eta_k)(\beta_{k-1} - \beta_k) - \eta_k \beta_k\big) \frac{1}{2} \|y_{\beta_k}^\star(Ax_k)\|^2 + \eta_k \|\nabla_x \mathbb{E}_\Omega f(x_k, \omega) - d_k\| D_{\mathcal{X}}.$$

Let us choose $\eta_k$ and $\beta_k$ in a way to vanish the last term. By choosing $\eta_k = \frac{9}{k+8}$ and $\beta_k = \frac{\beta_0}{(k+8)^{\frac{1}{2}}}$ for $k \geq 1$ with some $\beta_0 > 0$, we get $(1 - \eta_k)(\beta_{k-1} - \beta_k) - \eta_k \beta_k < 0$. Hence, we end up with

$$F_{\beta_k}(x_{k+1}) - F^\star \leq (1 - \eta_k)\big(F_{\beta_{k-1}}(x_k) - F^\star\big) + \frac{\eta_k^2}{2} D_{\mathcal{X}}^2 (L_f + \frac{\|A\|^2}{\beta_k})(1 + \delta)$$

$$+ \eta_k \|\nabla_x \mathbb{E}_\Omega f(x_k, \omega) - d_k\| D_{\mathcal{X}}.$$

We now compute the expectation, use Jensen inequality and use Lemma 8 to obtain the final recursion:

$$\mathbb{E} F_{\beta_k}(x_{k+1}) - F^\star \leq (1 - \eta_k)\big(\mathbb{E} F_{\beta_{k-1}}(x_k) - F^\star\big) + \frac{\eta_k^2}{2} D_{\mathcal{X}}^2 (L_f + \frac{\|A\|^2}{\beta_k})(1 + \delta)$$

$$+ \eta_k \mathbb{E} \|\nabla_x \mathbb{E}_\Omega f(x_k, \omega) - d_k\| D_{\mathcal{X}}$$

$$\leq (1 - \eta_k)\big(\mathbb{E} F_{\beta_{k-1}}(x_k) - F^\star\big) + \frac{\eta_k^2}{2} D_{\mathcal{X}}^2 (L_f + \frac{\|A\|^2}{\beta_k})(1 + \delta)$$

$$+ \eta_k \sqrt{\mathbb{E} \|\nabla \mathbb{E}_\Omega f(x_k, \omega) - d_k\|^2} D_{\mathcal{X}}$$

$$\leq (1 - \eta_k)\big(\mathbb{E} F_{\beta_{k-1}}(x_k) - F^\star\big) + \frac{\eta_k^2}{2} D_{\mathcal{X}}^2 (L_f + \frac{\|A\|^2}{\beta_k})(1 + \delta)$$

$$+ \frac{9 D_{\mathcal{X}} \sqrt{Q}}{(k+8)^{\frac{4}{3}}}$$

Now, note that:

$$\frac{\eta_k^2}{2} D_\mathcal{X}^2 \left( L_f + \frac{\|A\|^2}{\beta_k} \right) = \frac{\eta_k^2}{2} D_\mathcal{X}^2 L_f + \frac{\eta_k^2}{2} D_\mathcal{X}^2 \frac{\|A\|^2}{\beta_k}$$

$$= \frac{\frac{81}{2}}{(k+8)^2} D_\mathcal{X}^2 L_f + \frac{\frac{81}{2}}{(k+8)^{\frac{3}{2}}} \beta_0 D_\mathcal{X}^2 \|A\|^2$$

$$\leq \frac{\frac{81}{2}}{(k+8)^{\frac{4}{3}}} D_\mathcal{X}^2 L_f + \frac{\frac{81}{2}}{(k+8)^{\frac{4}{3}}} \beta_0 D_\mathcal{X}^2 \|A\|^2$$

Therefore:

$$\mathbb{E}F_{\beta_k}(x_{k+1}) - F^\star \leq \left( 1 - \frac{9}{k+8} \right) \left( \mathbb{E}F_{\beta_{k-1}}(x_k) - F^\star \right)$$

$$+ \frac{\frac{81}{2} D_\mathcal{X}^2 (L_f + \beta_0 \|A\|^2)(1+\delta) + 9 D_\mathcal{X} \sqrt{Q}}{(k+8)^{\frac{4}{3}}}$$

For simplicity, let $C_\delta := \frac{81}{2} D_\mathcal{X}^2 (L_f + \beta_0 \|A\|^2)(1+\delta) + 9 D_\mathcal{X} \sqrt{Q}$ and $\mathcal{E}_{k+1} := \mathbb{E}F_{\beta_k}(x_{k+1}) - F^\star$. Then, we need to solve the following recursive equation:

$$\mathcal{E}_{k+1} \leq \left( 1 - \frac{9}{k+8} \right) \mathcal{E}_k + \frac{C_\delta}{(k+8)^{\frac{4}{3}}} \tag{34}$$

Let the induction hypothesis for $k \geq 1$ be:

$$\mathcal{E}_{k+1} \leq 9^{\frac{1}{3}} \frac{C_\delta}{(k+8)^{\frac{1}{3}}}$$

For the base case $k = 1$ we need to prove $\mathcal{E}_2 \leq C_\delta$. From Equation (34) we have $\mathcal{E}_2 \leq \frac{C_\delta}{(9)^{\frac{4}{3}}} < C_\delta$ as $9^{\frac{4}{3}} > 1$ Now:

$$\mathcal{E}_{k+1} \leq \left( 1 - \frac{9}{k+8} \right) \mathcal{E}_k + \frac{C_\delta}{(k+8)^{\frac{4}{3}}}$$

$$\leq \left( 1 - \frac{9}{k+8} \right) 9^{\frac{1}{3}} \frac{C_\delta}{(k+7)^{\frac{1}{3}}} + \frac{C_\delta}{(k+8)^{\frac{4}{3}}}$$

$$\leq \left( 1 - \frac{9}{k+8} \right) 9^{\frac{1}{3}} \frac{C_\delta}{(k+7)^{\frac{1}{3}}} + 9^{\frac{1}{3}} \frac{C_\delta}{(k+8)^{\frac{4}{3}}}$$

$$\leq \left( 1 - \frac{9}{k+8} \right) 9^{\frac{1}{3}} \frac{C_\delta}{(k+7)^{\frac{1}{3}}} + 9^{\frac{1}{3}} \frac{C_\delta}{(k+7)^{\frac{1}{3}}(k+8)}$$

$$= 9^{\frac{1}{3}} \frac{C_\delta}{(k+8)(k+7)^{\frac{1}{3}}}(k-1)$$

$$\leq 9^{\frac{1}{3}} \frac{C_\delta}{(k+8)^{\frac{1}{3}}}$$

$\square$

**Corollary' 1.** *Assume that $g : \mathbb{R}^d \to \mathbb{R}$ is $L_g$-Lipschitz continuous. Then, the sequence $x_k$ generated by Algorithm 1 satisfies the following convergence bound for $k \geq 1$:*

$$\mathbb{E}F(x_{k+1}) - F^\star \leq 9^{\frac{1}{3}} \frac{C_\delta}{(k+8)^{\frac{1}{3}}} + \frac{\beta_0 L_g^2}{2\sqrt{k+8}}.$$

*Proof.* The proof is trivial and the technique comes from [41]. We report it for completeness. If $g : \mathbb{R}^d \to \mathbb{R} \cup \{+\infty\}$ is $L_g$-Lipschitz continuous from equation (2.7) in [32] and the duality between Lipshitzness and bounded support (*cf.* Lemma 5 in [5]) we have:

$$g_\beta(z) \leq g(z) \leq g_\beta(z) + \frac{\beta}{2} L_g^2 \tag{35}$$

Using this fact, we write:

$$g(Ax_{k+1}) \leq g_{\beta_k}(Ax_{k+1}) + \frac{\beta_k L_g^2}{2}$$
$$= g_{\beta_k}(Ax_{k+1}) + \frac{\beta_0 L_g^2}{2\sqrt{k+8}}.$$

We complete the proof by adding $\mathbb{E}\mathbb{E}_\Omega f(x_{k+1}, \omega) - F^\star$ to both sides:

$$\mathbb{E}F(x_{k+1}) - F^\star \leq \mathbb{E}F_{\beta_k}(x_{k+1}) - F^\star + \frac{\beta_0 L_g^2}{2\sqrt{k+8}}$$
$$\leq 9^{\frac{1}{3}} \frac{C_\delta}{(k+8)^{\frac{1}{3}}} + \frac{\beta_0 L_g^2}{2\sqrt{k+8}}.$$

$\square$

**Corollary' 2.** *Assume that $g : \mathbb{R}^d \to \mathbb{R}$ is the indicator function of a simple convex set $\mathcal{K}$. Then, the sequence $x_k$ generated by Algorithm 1 satisfies:*

$$\mathbb{E}\mathbb{E}_\Omega f(x_k, \omega) - f^\star \geq -\|y^\star\| \, \mathbb{E}\text{dist}(Ax_k, \mathcal{K})$$
$$\mathbb{E}\mathbb{E}_\Omega f(x_k, \omega) - f^\star \leq 9^{\frac{1}{3}} \frac{C_\delta}{(k+8)^{\frac{1}{3}}}$$
$$\mathbb{E}\text{dist}(Ax_k, \mathcal{K}) \leq \frac{2\beta_0\|y^\star\|}{\sqrt{k+8}} + \frac{2\sqrt{2 \cdot 9^{\frac{1}{3}}C_\delta\beta_0}}{(k+8)^{\frac{5}{12}}}$$

*Proof.* We adapt to our rate the proof technique of Theorem 4.3 in [41]. From the Lagrange saddle point theory, we know that the following bound holds $\forall x \in \mathcal{X}$ and $\forall r \in \mathcal{K}$:

$$f^\star \leq \mathcal{L}(x, r, y^\star) = \mathbb{E}_\Omega f(x, \omega) + \langle y_\star, \, Ax - r \rangle$$
$$\leq \mathbb{E}_\Omega f(x, \omega) + \|y_\star\| \|Ax - r\|,$$

Since $x_{k+1} \in \mathcal{X}$ and taking the expectation, we get

$$\mathbb{E}\mathbb{E}_\Omega f(x_{k+1}, \omega) - f^\star \geq -\mathbb{E} \min_{r \in \mathcal{K}} \|y^\star\| \|Ax_{k+1} - r\|$$
$$= -\|y^\star\| \mathbb{E}\text{dist}(Ax_{k+1}, \mathcal{K}). \qquad (36)$$

This proves the first bound in Corollary 2.

The second bound directly follows by Theorem 9 as

$$\mathbb{E}\mathbb{E}_\Omega f(x_{k+1}, \omega) - f^\star \mathbb{E} \leq \mathbb{E}\mathbb{E}_\Omega f(x_{k+1}, \omega) - f^\star + \frac{1}{2\beta_k}\mathbb{E}\left[\text{dist}(Ax_{k+1}, \mathcal{K})\right]^2$$
$$\leq \mathbb{E}F_{\beta_k}(x_{k+1}) - F^\star$$
$$\leq 9^{\frac{1}{3}} \frac{C_\delta}{(k+8)^{\frac{1}{3}}}.$$

Now, we combine this with (36), and we get

$$-\|y^\star\|\mathbb{E}\text{dist}(Ax_{k+1}, \mathcal{K}) + \frac{1}{2\beta_k}\mathbb{E}\left[\text{dist}(Ax_{k+1}, \mathcal{K})\right]^2 \leq 9^{\frac{1}{3}} \frac{C_\delta}{(k+8)^{\frac{1}{3}}}$$

This is a second order inequality in terms of $\mathbb{E}\text{dist}(Ax_k, \mathcal{K})$. Solving this inequality, we get

$$\mathbb{E}\text{dist}(Ax_{k+1}, \mathcal{K}) \leq \frac{2\beta_0\|y^\star\|}{\sqrt{k+8}} + \frac{2\sqrt{2 \cdot 9^{\frac{1}{3}}C_\delta\beta_0}}{(k+8)^{\frac{5}{12}}}.$$

$\square$

## D   Inexact Oracle with Multiplicative Error

**Lemma 10.** *For any given iteration $k \geq 1$ of Algorithm 1 the following relation holds:*

$$\langle \nabla F_{\beta_k}(x_k),\, \tilde{s}_k - x_k \rangle \leq \|\nabla_x \mathbb{E}_\Omega f(x_k, \omega) - d_k\| D_{\mathcal{X}}$$
$$+ \delta \left[ f^\star - \mathbb{E}_\Omega f(x_k, \omega) + g(Ax^\star) - g_{\beta_k}(Ax_k) - \frac{\beta_k}{2}\|y_{\beta_k}^*(Ax_k)\|^2 \right]$$

*where $\delta \in (0, 1]$ is the accuracy of the inexact lmo with multiplicative error.*

*Proof.*

$$\langle \nabla F_{\beta_k}(x_k),\, \tilde{s}_k - x_k \rangle = \langle \nabla_x \mathbb{E}_\Omega f(x_k, \omega),\, \tilde{s}_k - x_k \rangle + \langle A^\top \nabla g_{\beta_k}(Ax_k),\, \tilde{s}_k - x_k \rangle$$
$$= \langle \nabla_x \mathbb{E}_\Omega f(x_k, \omega),\, \tilde{s}_k - x_k \rangle + \langle A^\top \nabla g_{\beta_k}(Ax_k),\, \tilde{s}_k - x_k \rangle + \langle d_k,\, \tilde{s}_k - x_k \rangle - \langle d_k,\, \tilde{s}_k - x_k \rangle$$
$$= \langle \nabla_x \mathbb{E}_\Omega f(x_k, \omega) - d_k,\, \tilde{s}_k - x_k \rangle + \langle d_k + A^\top \nabla g_{\beta_k}(Ax_k),\, \tilde{s}_k - x_k \rangle$$
$$\leq \langle \nabla_x \mathbb{E}_\Omega f(x_k, \omega) - d_k,\, \tilde{s}_k - x_k \rangle + \delta \langle d_k + A^\top \nabla g_{\beta_k}(Ax_k),\, s_k - x_k \rangle \qquad (37)$$
$$\leq \langle \nabla_x \mathbb{E}_\Omega f(x_k, \omega) - d_k,\, \tilde{s}_k - x_k \rangle + \delta \langle d_k + A^\top \nabla g_{\beta_k}(Ax_k),\, x^\star - x_k \rangle \qquad (38)$$
$$= \langle \nabla_x \mathbb{E}_\Omega f(x_k, \omega) - d_k,\, \tilde{s}_k - x_k \rangle + \delta \langle d_k + A^\top \nabla g_{\beta_k}(Ax_k),\, x^\star - x_k \rangle$$
$$+ \delta \langle \nabla_x \mathbb{E}_\Omega f(x_k, \omega),\, x^\star - x_k \rangle - \delta \langle \nabla_x \mathbb{E}_\Omega f(x_k, \omega),\, x^\star - x_k \rangle$$
$$= \langle \nabla_x \mathbb{E}_\Omega f(x_k, \omega) - d_k,\, \tilde{s}_k - x_k - \delta x^\star + \delta x_k \rangle$$
$$+ \delta \langle \nabla_x \mathbb{E}_\Omega f(x_k, \omega) + A^\top \nabla g_{\beta_k}(Ax_k),\, x^\star - x_k \rangle$$
$$\leq \|\nabla_x \mathbb{E}_\Omega f(x_k, \omega) - d_k\| \|\tilde{s}_k - ((1 - \delta)x_k + \delta x^\star)\|$$
$$+ \delta \langle \nabla_x \mathbb{E}_\Omega f(x_k, \omega) + A^\top \nabla g_{\beta_k}(Ax_k),\, x^\star - x_k \rangle \qquad (39)$$
$$\leq \|\nabla_x \mathbb{E}_\Omega f(x_k, \omega) - d_k\| D_{\mathcal{X}} + \delta \langle \nabla_x \mathbb{E}_\Omega f(x_k, \omega) + A^\top \nabla g_{\beta_k}(Ax_k),\, x^\star - x_k \rangle \qquad (40)$$

where the Equation (37) is the definition of inexact oracle with multiplicative error, Equation (38) is because $s_k$ is a solution of $\min_{x \in \mathcal{X}} \langle d_k + A^\top \nabla g_{\beta_k}(Ax_k),\, x \rangle$, Equation (39) is cauchy-schwarz and Equation (40) is the diameter definition noting that $(1 - \delta)x_k + \delta x^\star \in \mathcal{X}$ as it is a convex combination of elements in $\mathcal{X}$.

Now, convexity of $\mathbb{E}_\Omega f(x_k, \omega)$ ensures $\langle \nabla \mathbb{E}_\Omega f(x_k, \omega),\, x^\star - x_k \rangle \leq f^\star - \mathbb{E}_\Omega f(x_k, \omega)$. Using property (29), we have

$$\langle A^\top \nabla g_{\beta_k}(Ax_k),\, x^\star - x_k \rangle = \langle \nabla g_{\beta_k}(Ax_k),\, Ax^\star - Ax_k \rangle$$
$$\leq g(Ax^\star) - g_{\beta_k}(Ax_k) - \frac{\beta_k}{2}\|y_{\beta_k}^*(Ax_k)\|^2.$$

Therefore:

$$\langle \nabla F_{\beta_k}(x_k),\, \tilde{s}_k - x_k \rangle \leq \|\nabla_x \mathbb{E}_\Omega f(x_k, \omega) - d_k\| D_{\mathcal{X}}$$
$$+ \delta \left[ f^\star - \mathbb{E}_\Omega f(x_k, \omega) + g(Ax^\star) - g_{\beta_k}(Ax_k) - \frac{\beta_k}{2}\|y_{\beta_k}^*(Ax_k)\|^2 \right]$$

$\square$

**Lemma 11.** *For any $k \geq 1$ the estimate of the gradient computed in Algorithm 1 satisfies:*

$$\mathbb{E}\left[ \|\nabla_x \mathbb{E}_\Omega f(x_k, \omega) - d_k\|^2 \right] \leq \frac{Q}{(\delta(k-1) + 9)^{\frac{2}{3}}}$$

*where $Q = \max \left\{ \|\nabla_x \mathbb{E}_\Omega f(x_1, \omega) - d_1\|^2 7^{\frac{2}{3}},\, 16\sigma^2 + 81 L_f^2 D_{\mathcal{X}}^2 \right\}$*

*Proof.* This lemma simply applies Lemma 1 and Lemma 17 of [31] to our different stepsizes. We report all the steps for clarity and completeness. First, we invoke Lemma 1:

$$\mathbb{E}\left[\|\nabla_x\mathbb{E}_\Omega f(x_k,\omega) - d_k\|^2\right] \leq \left(1 - \frac{\rho_k}{2}\right)\|\nabla_x\mathbb{E}_\Omega f(x_{k-1},\omega) - d_{k-1}\|^2 + \rho_k^2\sigma^2 + \frac{2L_f^2 D_{\mathcal{X}}^2 \eta_{k-1}^2}{\rho_k}$$

$$\leq \left(1 - \frac{2}{(\delta(k-2)+9)^{\frac{2}{3}}}\right)\|\nabla_x\mathbb{E}_\Omega f(x_{k-1},\omega) - d_{k-1}\|^2 + \frac{16\sigma^2 + 81L_f^2 D_{\mathcal{X}}^2}{(\delta(k-2)+9)^{\frac{4}{3}}} \qquad (41)$$

where we used $\rho_k = \frac{4}{(\delta(k-2)+9)^{\frac{2}{3}}}$. Now, Lemma 17 of [31] gives the following solution:

$$\phi_t \leq \frac{Q}{(k+k_0+1)^\alpha}$$

to the recursion

$$\phi_k \leq \left(1 - \frac{c}{(k+k_0)^\alpha}\right)\phi_{k-1} + \frac{b}{(k+k_0)^{2\alpha}}$$

where $b \geq 0$, $c > 1$, $\alpha \leq 1$, $k_0 \geq 0$ and $\tilde{Q} := \max\{\phi_1 k_0^\alpha, b/(c-1)\}$ Applying this lemma to Equation (41) with $k_0 = \frac{9}{\delta} - 2$, $\alpha = \frac{2}{3}$, $c = \frac{2}{\delta^{\frac{2}{3}}}$, $b = \frac{16\sigma^2 + 81L_f^2 D_{\mathcal{X}}^2}{\delta^{\frac{4}{3}}}$ gives:

$$\mathbb{E}\left[\|\nabla_x\mathbb{E}_\Omega f(x_k,\omega) - d_k\|^2\right] \leq \frac{Q}{(\delta(k-1)+9)^{\frac{2}{3}}}$$

where $Q = \max\left\{\|\nabla_x\mathbb{E}_\Omega f(x_1,\omega) - d_1\|^2(9-2\delta)^{\frac{2}{3}}, \frac{16\sigma^2 + 81L_f^2 D_{\mathcal{X}}^2}{(2-\delta^{\frac{2}{3}})}\right\}$ $\qquad\square$

**Theorem 12.** *The sequence $x_k$ generated by Algorithm 1 with approximate lmo of the form* (24), *and modifying $\eta_k = \frac{9}{\delta(k-1)+9}$, $\beta_k = \frac{\beta_0}{\sqrt{\delta(k-1)+9}}$ and $\rho_k = \frac{4}{(\delta(k-2)+9)^{\frac{2}{3}}}$ satisfies:*

$$\mathbb{E}F_{\beta_k}(x_{k+1}) - F^\star \leq 9^{\frac{1}{3}}\frac{\frac{C}{\delta} + \mathcal{E}_1}{(\delta(k-1)+9)^{\frac{1}{3}}}$$

*where $\mathcal{E}_1 := F_{\frac{\beta_0}{\sqrt{9}}}(x_1) - F^\star$.*

*Proof.* First, we use the smoothness of $F_{\beta_k}$ to upper bound the progress. Note that $F_{\beta_k}$ is $(L_f + \|A\|^2/\beta_k)$-smooth.

$$F_{\beta_k}(x_{k+1}) \leq F_{\beta_k}(x_k) + \eta_k\langle\nabla F_{\beta_k}(x_k), \tilde{s}_k - x_k\rangle + \frac{\eta_k^2}{2}\|\tilde{s}_k - x_k\|^2(L_f + \frac{\|A\|^2}{\beta_k})$$

$$\leq F_{\beta_k}(x_k) + \eta_k\langle\nabla F_{\beta_k}(x_k), \tilde{s}_k - x_k\rangle + \frac{\eta_k^2}{2}D_{\mathcal{X}}^2(L_f + \frac{\|A\|^2}{\beta_k}), \qquad (42)$$

where $\tilde{s}_k$ denotes the atom selected by the approximate *lmo* with multiplicative accuracy, and the second inequality follows since $\tilde{s}_k \in \mathcal{X}$.

Using Lemma 10 we get:

$$F_{\beta_k}(x_{k+1}) \leq F_{\beta_k}(x_k) + \eta_k\delta\left(f^\star - \mathbb{E}_\Omega f(x_k,\omega) + g(Ax^\star) - g_{\beta_k}(Ax_k) - \frac{\beta_k}{2}\|\nabla y_{\beta_k}^*(Ax_k)\|^2\right)$$

$$+ \frac{\eta_k^2}{2}D_{\mathcal{X}}^2(L_f + \frac{\|A\|^2}{\beta_k}) + \eta_k\|\nabla_x\mathbb{E}_\Omega f(x_k,\omega) - d_k\|D_{\mathcal{X}} \qquad (43)$$

$$= (1 - \delta\eta_k)F_{\beta_k}(x_k) + \delta\eta_k F^\star - \frac{\delta\eta_k\beta_k}{2}\|\nabla y_{\beta_k}^*(Ax_k)\|^2$$

$$+ \frac{\eta_k^2}{2}D_{\mathcal{X}}^2(L_f + \frac{\|A\|^2}{\beta_k}) + \eta_k\|\nabla_x\mathbb{E}_\Omega f(x_k,\omega) - d_k\|D_{\mathcal{X}}.$$

Now, using (33), we get

$$F_{\beta_k}(x_k) = \mathbb{E}_\Omega f(x_k, \omega) + g_{\beta_k}(Ax_k)$$

$$\leq \mathbb{E}_\Omega f(x_k, \omega) + g_{\beta_{k-1}}(Ax_k) + \frac{\beta_{k-1} - \beta_k}{2} \|y^*_{\beta_k}(Ax_k)\|^2$$

$$= F_{\beta_{k-1}}(x_k) + \frac{\beta_{k-1} - \beta_k}{2} \|y^*_{\beta_k}(Ax_k)\|^2.$$

We combine this with (43) and subtract $F^\star$ from both sides to get

$$F_{\beta_k}(x_{k+1}) - F^\star \leq (1 - \delta\eta_k)\big(F_{\beta_{k-1}}(x_k) - F^\star\big) + \frac{\eta_k^2}{2} D_\mathcal{X}^2 (L_f + \frac{\|A\|^2}{\beta_k})$$

$$+ \big((1 - \delta\eta_k)(\beta_{k-1} - \beta_k) - \delta\eta_k\beta_k\big)\frac{1}{2}\|y^*_{\beta_k}(Ax_k)\|^2 + \eta_k\|\nabla_x\mathbb{E}_\Omega f(x_k, \omega) - d_k\|D_\mathcal{X}.$$

Let us choose $\eta_k$ and $\beta_k$ in a way to vanish the last term. By choosing $\eta_k = \frac{9}{\delta(k-1)+9}$ and $\beta_k = \frac{\beta_0}{(\delta(k-1)+9)^{\frac{1}{2}}}$ for $k \geq 1$ with some $\beta_0 > 0$, we get $(1 - \delta\eta_k)(\beta_{k-1} - \beta_k) - \delta\eta_k\beta_k < 0$. Hence, we end up with

$$F_{\beta_k}(x_{k+1}) - F^\star \leq (1 - \delta\eta_k)\big(F_{\beta_{k-1}}(x_k) - F^\star\big) + \frac{\eta_k^2}{2} D_\mathcal{X}^2 (L_f + \frac{\|A\|^2}{\beta_k}) + \eta_k\|\nabla_x\mathbb{E}_\Omega f(x_k, \omega) - d_k\|D_\mathcal{X}.$$

We now compute the expectation, use Jensen inequality and use Lemma 8 to obtain the final recursion:

$$\mathbb{E}F_{\beta_k}(x_{k+1}) - F^\star \leq (1 - \delta\eta_k)\big(\mathbb{E}F_{\beta_{k-1}}(x_k) - F^\star\big) + \frac{\eta_k^2}{2} D_\mathcal{X}^2 (L_f + \frac{\|A\|^2}{\beta_k})$$

$$+ \eta_k \mathbb{E}\|\nabla_x\mathbb{E}_\Omega f(x_k, \omega) - d_k\|D_\mathcal{X}$$

$$\leq (1 - \delta\eta_k)\big(\mathbb{E}F_{\beta_{k-1}}(x_k) - F^\star\big) + \frac{\eta_k^2}{2} D_\mathcal{X}^2 (L_f + \frac{\|A\|^2}{\beta_k})$$

$$+ \eta_k \sqrt{\mathbb{E}\|\nabla_x\mathbb{E}_\Omega f(x_k, \omega) - d_k\|^2} D_\mathcal{X}$$

$$\leq (1 - \delta\eta_k)\big(\mathbb{E}F_{\beta_{k-1}}(x_k) - F^\star\big) + \frac{\eta_k^2}{2} D_\mathcal{X}^2 (L_f + \frac{\|A\|^2}{\beta_k}) + \frac{9 D_\mathcal{X}\sqrt{Q}}{(\delta(k-1)+9)^{\frac{4}{3}}}$$

Now, note that:

$$\frac{\eta_k^2}{2} D_\mathcal{X}^2 (L_f + \frac{\|A\|^2}{\beta_k}) = \frac{\eta_k^2}{2} D_\mathcal{X}^2 L_f + \frac{\eta_k^2}{2} D_\mathcal{X}^2 \frac{\|A\|^2}{\beta_k}$$

$$= \frac{81/2}{(\delta(k-1)+9)^2} D_\mathcal{X}^2 L_f + \frac{81/2}{(\delta(k-1)+9)^{\frac{3}{2}}} \beta_0 D_\mathcal{X}^2 \|A\|^2$$

$$\leq \frac{81/2}{(\delta(k-1)+9)^{\frac{4}{3}}} D_\mathcal{X}^2 L_f + \frac{81/2}{(\delta(k-1)+9)^{\frac{4}{3}}} \beta_0 D_\mathcal{X}^2 \|A\|^2$$

Therefore:

$$\mathbb{E}F_{\beta_k}(x_{k+1}) - F^\star \leq \left(1 - \frac{9\delta}{\delta(k-1)+9}\right)\big(F_{\beta_{k-1}}(x_k) - F^\star\big) + \frac{\frac{81}{2}D_\mathcal{X}^2(L_f + \beta_0\|A\|^2) + 9D_\mathcal{X}\sqrt{Q}}{(\delta(k-1)+9)^{\frac{4}{3}}}$$

For simplicity, let $C := \frac{81}{2} D_\mathcal{X}^2 (L_f + \beta_0\|A\|^2) + 9 D_\mathcal{X}\sqrt{Q}$ and $\mathcal{E}_{k+1} := \mathbb{E}F_{\beta_k}(x_{k+1}) - F^\star$. Then, we need to solve the following recursive equation:

$$\mathcal{E}_{k+1} \leq \left(1 - \frac{9\delta}{\delta(k-1)+9}\right)\mathcal{E}_k + \frac{C}{(\delta(k-1)+9)^{\frac{4}{3}}} \tag{44}$$

Let the induction hypothesis for $k \geq 1$ be:

$$\mathcal{E}_{k+1} \leq 9^{\frac{1}{3}} \frac{\frac{C}{\delta} + \mathcal{E}_1}{(\delta(k-1)+9)^{\frac{1}{3}}}$$

The base case $k = 1$ is trivial as from (44) we have $\mathcal{E}_2 \leq (1 - \delta)\,\mathcal{E}_1 + \frac{C}{9^{\frac{4}{3}}} \leq \mathcal{E}_1 + \frac{C}{9^{\frac{4}{3}}} \leq \mathcal{E}_1 + \frac{C}{\delta}$

For simplicity let $K := \delta(k-1) + 9$. From Equation (34) we have $\mathcal{E}_2 \leq \frac{C}{(9)^{\frac{4}{3}}} < C$ as $9^{\frac{4}{3}} > 1$ Now:

$$\mathcal{E}_{k+1} \leq \left(1 - \frac{9\delta}{K}\right)\mathcal{E}_k + \frac{C}{(K)^{\frac{4}{3}}}$$

$$\leq \left(1 - \frac{9\delta}{K}\right)9^{\frac{1}{3}}\frac{\frac{C}{\delta} + \mathcal{E}_1}{(K - \delta)^{\frac{1}{3}}} + \frac{C}{(K)^{\frac{4}{3}}}$$

$$\leq \left(1 - \frac{9\delta}{K}\right)9^{\frac{1}{3}}\frac{\frac{C}{\delta} + \mathcal{E}_1}{(K - \delta)^{\frac{1}{3}}} + 9^{\frac{1}{3}}\delta\frac{\frac{C}{\delta} + \mathcal{E}_1}{K(K - \delta)^{\frac{1}{3}}}$$

$$\leq \left(1 - \frac{8\delta}{K}\right)9^{\frac{1}{3}}\frac{\frac{C}{\delta} + \mathcal{E}_1}{(K - \delta)^{\frac{1}{3}}}$$

$$\leq 9^{\frac{1}{3}}\frac{\frac{C}{\delta} + \mathcal{E}_1}{(K + \delta)^{\frac{1}{3}}}$$

$\square$

**Corollary' 5.** *Assume that $g$ is $L_g$-Lipschitz continuous. Then, the sequence $x_k$ generated by Algorithm 1 with approximate lmo (24), and modifying $\eta_k = \frac{9}{\delta(k-1)+9}$, $\beta_k = \frac{\beta_0}{\sqrt{\delta(k-1)+9}}$ and $\rho_k = \frac{4}{(\delta(k-2)+9)^{\frac{2}{3}}}$ satisfies:*

$$\mathbb{E}F(x_{k+1}) - F^\star \leq 9^{\frac{1}{3}}\frac{\frac{C}{\delta} + \mathcal{E}_1}{(\delta(k-1)+9)^{\frac{1}{3}}} + \frac{\beta_0 L_g^2}{2\sqrt{\delta(k-1)+9}},$$

*We can optimize $\beta_0$ from this bound if $\delta$ is known.*

*Proof.* If $g : \mathbb{R}^d \to \mathbb{R} \cup \{+\infty\}$ is $L_g$-Lipschitz continuous we get from (35):

$$g(Ax_{k+1}) \leq g_{\beta_k}(Ax_{k+1}) + \frac{\beta_k L_g^2}{2}$$

$$= g_{\beta_k}(Ax_{k+1}) + \frac{\beta_0 L_g^2}{2\sqrt{\delta(k-1)+9}}.$$

We complete the proof by adding $\mathbb{E}\mathbb{E}_\Omega f(x_{k+1}, \omega) - F^\star$ to both sides:

$$\mathbb{E}F(x_{k+1}, \omega) - F^\star \leq \mathbb{E}F_{\beta_k}(x_{k+1}) - F^\star + \frac{\beta_0 L_g^2}{2\sqrt{k+8}}$$

$$\leq 9^{\frac{1}{3}}\frac{\frac{C}{\delta} + \mathcal{E}_1}{(\delta(k-1)+9)^{\frac{1}{3}}} + \frac{\beta_0 L_g^2}{2\sqrt{\delta(k-1)+9}}.$$

$\square$

**Corollary' 6.** *Assume that $g$ is the indicator function of a simple convex set $\mathcal{K}$. Then, the sequence $x_k$ generated by Algorithm 1 with approximate lmo (24), and modifying $\eta_k = \frac{9}{\delta(k-1)+9}$, $\beta_k = \frac{\beta_0}{\sqrt{\delta(k-1)+9}}$ and $\rho_k = \frac{4}{(\delta(k-2)+9)^{\frac{2}{3}}}$ satisfies:*

$$\mathbb{E}\mathbb{E}_\Omega f(x_{k+1}, \omega) - f^\star \geq -\|y^\star\|\,\mathbb{E}\text{dist}(Ax_{k+1}, \mathcal{K})$$

$$\mathbb{E}\mathbb{E}_\Omega f(x_{k+1}, \omega) - f^\star \leq 9^{\frac{1}{3}}\frac{\frac{C}{\delta} + \mathcal{E}_1}{(\delta(k-1)+9)^{\frac{1}{3}}}$$

$$\mathbb{E}\text{dist}(Ax_{k+1}, \mathcal{K}) \leq \frac{2\beta_0\|y^\star\|}{\sqrt{\delta(k-1)+9}} + \frac{2\sqrt{2 \cdot 9^{\frac{1}{3}}(\frac{C}{\delta} + \mathcal{E}_1)\beta_0}}{(\delta(k-1)+9)^{\frac{5}{12}}}$$

*Proof.* We adapt to our rate the proof technique of Theorem 4.3 in [41]. From the Lagrange saddle point theory, we know that the following bound holds $\forall x \in \mathcal{X}$ and $\forall r \in \mathcal{K}$:

$$f^\star \leq \mathcal{L}(x, r, y^\star) = \mathbb{E}_\Omega f(x, \omega) + \langle y_\star, Ax - r \rangle$$
$$\leq \mathbb{E}_\Omega f(x, \omega) + \|y_\star\| \|Ax - r\|,$$

Since $x_{k+1} \in \mathcal{X}$, we get after taking the expectation

$$\mathbb{E}\mathbb{E}_\Omega f(x_{k+1}, \omega) - f^\star \geq -\mathbb{E} \min_{r \in \mathcal{K}} \|y^\star\| \|Ax_{k+1} - r\|$$
$$= -\|y^\star\| \mathbb{E}\mathrm{dist}(Ax_{k+1}, \mathcal{K}). \tag{45}$$

This proves the first bound in Corollary 2.

The second bound directly follows by Theorem 9 as

$$\mathbb{E}\mathbb{E}_\Omega f(x_{k+1}, \omega) - f^\star \leq \mathbb{E}\mathbb{E}_\Omega f(x_{k+1}, \omega) - f^\star + \frac{1}{2\beta_k} \mathbb{E}\left[\mathrm{dist}(Ax_{k+1}, \mathcal{K})\right]^2$$
$$\leq \mathbb{E} F_{\beta_k}(x_{k+1}) - F^\star$$
$$\leq 9^{\frac{1}{3}} \frac{\frac{C}{\delta} + \mathcal{E}_1}{(\delta(k-1) + 9)^{\frac{1}{3}}}.$$

Now, we combine this with (45), and we get

$$-\|y^\star\| \mathbb{E}\mathrm{dist}(Ax_{k+1}, \mathcal{K}) + \frac{1}{2\beta_k} \mathbb{E}\left[\mathrm{dist}(Ax_{k+1}, \mathcal{K})\right]^2 \leq 9^{\frac{1}{3}} \frac{\frac{C}{\delta} + \mathcal{E}_1}{(\delta(k-1) + 9)^{\frac{1}{3}}}$$

This is a second order inequality in terms of $\mathbb{E}\mathrm{dist}(Ax_k, \mathcal{K})$. Solving this inequality, we get

$$\mathbb{E}\mathrm{dist}(Ax_{k+1}, \mathcal{K}) \leq \frac{2\beta_0 \|y^\star\|}{\sqrt{\delta(k-1) + 9}} + \frac{2\sqrt{2 \cdot 9^{\frac{1}{3}} (\frac{C}{\delta} + \mathcal{E}_1)\beta_0}}{(\delta(k-1) + 9)^{\frac{5}{12}}}.$$

$\square$