[Reviews · NeurIPS 2019]

Reviewer 1



**** Post rebuttal I have read the rebuttal and other review, and I keep my initial score. **** The paper present a stochastic Frank-Wolfe method that solves min_{x} E_W f(x,w) + g(Ax), where f(.,w) is a convex, smooth function and g(.) is convex and potentially nonsmooth. The paper uses the Nesterov's smoothing technique to deal with the nonsmooth component. Shortly, it approximates the nonsmooth function g(.) by g_beta, where beta controls the smoothness. The idea is to decrease beta as a function of the iteration counter so that g_beta becomes closer to g. The paper goes straight to the point and explains clearly the setting, the algorithm, and the technique used to design it. The final idea is elegant and leads to a quite simple algorithm. Moreover, the numerical experiments are well explained and are convincing, as the authors tested their approach to many different problems. In term of novelty, the analysis seems pretty classical to me. The first step is the analysis of stochastic Frank-Wolfe for smooth problems, and then the second step uses Nesterov's results on smoothing techniques. However, this simple combination still leads to an interesting result. A possible improvement for clarity may be to detail a bit more the proof of theorem 1, even if the sketch is already sufficient to be convinced that the claim is valid. I think this is a good, well-written paper that presents a good contribution. However, even if the authors explained clearly their contribution and existing work on this topic, I may not have the right background to judge how different this algorithm is from other existing approaches. Therefore, I recommend acceptance of this paper - but I may change my opinion depending on the other reviews

Reviewer 2



Originality. This paper is a good combination of the techniques in [28] and [36]. This allows to obtain a stochastic method with cheaper iteration for the problem considered in [36]. At the same time, Table 1 does not give clear advantage in comparison with [36]. It seems that even for moderate eps, 1/eps^3 can be larger than the size N of the dataset. Quality. As far as I see, the proofs are correct. Experiments show the advantage of the stochastic method. There are some misprints and questions. I refer to the supplementary material. 1. line 41 form -> from 2. line 145 computetion -> computation 3. Lemma 7. There should be \tilde{s}_k in the l.h.s. of the inequality instead of s_k. Similarly in the proof starting with the first inequality in the long derivation, tilde is missing for s_k in the first term $<\nabla_x E_{\Omega} f(x_k, \Omega)-d_k, s_k-x_k >$. 4. In (29) where does the first inequality follow from? 5. Line 505 cauchy-schwarz -> Cauchy-Schwarz 6. Line 532. Could you please give more details on how this inequality follows form the choice of the coefficients? Clarity. Except the questions listed above, the clarity is rather good. I don't have any further suggestions for improvement. Significance. Despite not optimal complexity, this method can be useful in practice because of its cheaper iteration and ability to solve the problem in the online setting. Also there is a room for follow-up papers with improved complexity. =========After rebuttal============ I'd like to thank the authors for their rebuttal. Providing more details in the final version will improve the quality of the paper. I leave my score unchanged.

Reviewer 3



Clarity and quality: The paper is easy to understand and the results are clearly stated and well-organized. I would like to suggest the authors move the contributions to the Introduction section. By the way, I prefer to remove the last contribution regarding the experiments, which I do not think this could be a contribution. The authors conducted several experiments, which is appreciated. However, it seems to me that the matrix completion problem does not satisfy convexity assumption. Significance: This paper considers a very interesting and widely used constrained convex optimization problem semidefinite program (SDP), which is to minimize a convex function over the positive semidefinite cone subject to some affine constraints. However, given existing results, the main theoretical contributions of this paper are about solving the problem (3), which could be limited. For example, both iteration complexity and sample complexity in [2] are better than this work, while [2] can be also used to solve (3). Originality: This paper proposed the first Frank-Wolf type method with theoretical guarantee for solving the considered constrained optimization problem in the stochastic setting, although this idea is simple. So I think the originality is ok.

[Author Response · NeurIPS 2019]

We thank the reviewers for their comments and time. We are glad there is a positive consensus: The proposed method and the analysis are recognized as novel and practically interesting in some important problem settings. Our paper is found well-written.

For the sake of completeness, we will address some of the minor remarks by the reviewers below:

**R3:** *Advantages against [2].*

The arithmetic cost per iteration of [2] is cubic in $n$ (as it requires a full singular value decomposition in general). Also, your comment made us realize that we missed a key qualifier information in Table 1. $1/\varepsilon^2$ iteration complexity of [2] holds only when the objective function is strongly-convex (see Section 4 in [2]). We will clarify this.

Based on this clarification we hope that the reviewer will now reconsider our score even more positively.

**R3:** *Convexity in matrix completion.*

There might be a misunderstanding. We consider the conventional convex matrix completion template with least squares loss with an additional box constraints. See Section 2.3.3. in [DY] for a similar setup (regularized version in the deterministic setting).

[DY] D. Davis, W. Yin. "A Three-Operator Splitting Scheme and its Optimization Applications" arXiv:1504.01032v1

**R1:** *Adding proof sketch.*

As suggested by the reviewer, we can extend the proof sketch the help the readers navigating the proof.

**R2:** *Where does (29) follows from?*

It follows from Lemma 1 in [4]. We cite it in the previous sentence. We will clarify this in the text.

**R2:** *Details on the inequality on Line 532.*

There is a typo in line 532, thank you for pointing it. Our bound is missing the $\delta$ terms. The correct version of the bound is $(1 - \delta\eta_k)(\beta_{k-1} - \beta_k) - \delta\eta_k\beta_k < 0$. One can verify this bound by mathematical induction technique.

**R1,R2,R3:** *Other comments.*

We thank all reviewers for their constructive comments on the clarity and presentation. We will consider their suggestions while preparing the camera-ready.

[Meta-Review · NeurIPS 2019]

All reviewers agreed that this paper makes an interesting contribution to NeurIPS. Please make sure to include the promised changes in the camera-ready version.